



# Benchmark forward gravity schemes: the gravity field of a realistic lithosphere model WINTERC-G

Barend Cornelis (Bart) Root[1,5], Josef Sebera[2,3], Wolfgang Szwillus[3], Cedric Thieulot[4],
Zdeněk Martinec[5], and Javier Fullea[6,5]

[1]Delft University of Technology, Department of Space Engineering, Delft, the Netherlands
[2]Astronomical Institute of the Czech Academy of Sciences, Prague, Czech Republic
[3]Christian Albrechts University, Department of Geosciences, Kiel, Germany
[4]Utrecht University, , Utrecht, the Netherlands
[5]Dublin Insitute for Advanced Studies, School of Cosmic Physics, Dublin, Ireland
[6]Universidad Complutense de Madrid (UCM), Physics of the Earth and Astrophysics department, Madrid, Spain

**Correspondence:** B.C. Root (b.c.root@tudelft.nl)

**Abstract.** Several alternative gravity forward modelling methodologies and associated numerical codes with their own advantages and limitations are available for the Solid Earth community. With the upcoming state-of-the-art lithosphere density models and accurate global gravity field data sets it is vital to understand the opportunities and limitations of the various approaches. In this paper, we discuss the four widely used techniques: global spherical harmonics (GSH), tesseroid integration (TESS), triangle integration (TRI), and hexahedral integration (HEX). A constant density shell benchmark shows that all four codes can produce similar precise gravitational potential fields. Two additional shell tests were conducted with more complicated density structures: lateral varying density structures and a Moho density interface between crust and mantle. The differences between the four codes were all below 1.5 percent of the modeled gravity signal suitable for reproducing satellite-acquired gravity data. TESS and GSH produced the most similar potential fields (<0.3 percent).

To examine the usability of the forward modelling codes for realistic geological structures, we use the global lithosphere model WINTERC-G, that was constrained, among other data, by satellite gravity field data computed using a spectral forward modeling approach. This spectral code was benchmarked against the GSH and it was confirmed that both approaches produce similar gravity solution with negligible differences between them. In the comparison of the different WINTERC-G-based gravity solutions, again GSH and TESS performed best. Only short-wavelength noise is present between the spectral and tesseroid forward modelling approaches, likely related to the different way in which the spherical harmonic analysis of the varying boundaries of the mass layer is performed. The Spherical harmonic basis functions produces small differences compared to the tesseroid elements especially at sharp interfaces, which introduces mostly short-wavelength differences. Nevertheless, both approaches (GSH and TESS) result in accurate solutions of the potential field with reasonable computational resources. Differences below 0.5 percent are obtained, resulting in residuals of 0.076 mGal standard deviation at 250 km height.

The biggest issue for TRI is the characteristic pattern in the residuals that is related to the grid layout. Increasing the resolution and filtering allows for the removal of most of this erroneous pattern, but at the expense of higher computational loads with respect to the other codes. The other spatial forward modelling scheme HEX has more difficulty in reproducing





similar gravity field solutions compared to GSH and TESS. These particular approaches need to go to higher resolutions, resulting in enormous computation efforts. The hexahedron-based code performs less than optimal in the forward modelling of

the gravity signature, especially of a lateral varying density interface. Care must be taken with any forward modelling software as the approximation of the geometry of the WINTERC-G model may deteriorate the gravity field solution.

## 1   Introduction

Dedicated gravimetric satellite missions such as NASA's GRACE and ESA's GOCE missions have generated unprecedented views of the Earth's gravity field (*Pail et al.*, 2015). One of the latest global gravity field model, XGM2016 (*Pail et al.* ,

2018), depicts a detailed map of the gravity anomalies caused by density variations in the Earth's interior. Such variations provide information on the density distribution within the Earth with homogeneous (global) quality that can be used in joint inversion studies of the subsurface combining gravity data with petrological and seismological constraints (*Kaban et al.* , 2014), such like the latest global lithosphere and upper mantle model WINTERC-G (*Fullea et al.*, 2020). However, the different spatial parametrizations used in seismology and potential field studies raise the question of compatibility between the various

approaches to forward model the gravitational attraction of the density models.

The recent global model WINTERC-G (*Fullea et al.*, 2020) is based on the thermochemical approach LitMod, that has been used previously for forward calculation (*Afonso et al.*, 2008; *Fullea et al.*, 2009) and inversion in the regional context (*Afonso et al.*, 2013a, b, 2016). The global WINTERC-G model (*Fullea et al.*, 2020) is the result of a two-stage coupled inversion process, consisting of 1-D stage followed by a 3-D stage. In the first stage, the model is represented as independent

lithospheric/upper mantle columns that are laterally distributed on a spherical triangular grid (*Wang and Dahlen*, 1995). In this stage, the model has no explicit lateral structural information other than that contained in the geophysical observable (surface wave dispersion curves, surface elevation and heat flow). In the second stage, the model is rendered in 3-D as a collection of 13 layers with varying layer thickness and laterally varying density gradient (see also Appendix C). The gravity effect of the layered representation is calculated using a spherical harmonics approach and used to drive the linearised density inversion in

order to fit the gravity field signal from XGM2016 (*Fullea et al.*, 2020). The choice of forward modelling approach to represent the model's potential field is in principle arbitrary, and it is unclear what effect a different decision would have on the resulting density structure. Furthermore, for certain applications a user might want to change to a different parametrization, for example when WINTERC-G is used as a starting point for more detailed regional modelling. We are thus motivated by the results of *Fullea et al.* (2020) to benchmark the effect of parametrization/discretization on lithospheric gravity calculation. Ultimately

we are driven by the need to quantify the effect of different discretizations and numerical approaches used to represent the real distribution of the Earth's 3-D density distribution and its associated gravity field.

Forward gravity-field modelling discretizations can be classified as space-domain or spectral-domain (*Hirt and Kuhn*, 2014). The two classes differ in how a continuous mass distribution $\rho(Q) \times \Sigma(Q)$ is derived from the discrete numbers given in the density model. Such a continuous distribution is required to evaluate Newton's integral and determines the gravitational





potential $V$ outside the mass body $\Sigma$ at location (e.g., *Rummel et al.* (1988)):

$$V(P) = G \iiint\limits_{\Sigma} \frac{\rho(Q)}{\ell(P,Q)} \, \mathrm{d}\Sigma(Q) \,. \tag{1}$$

where $G$ is the universal gravitational constant, $\rho$ is the mass density distribution within the body $\Sigma$ and $\ell(P,Q)$ is the Euclidean distance between the computation point $P(r, \Omega)$ and the infinitesimal volume element $\mathrm{d}\Sigma(Q)$ at location $Q(r', \Omega')$. Space-domain forward modelling methods evaluate Newton's integral directly, where an arbitrary mass object is approximated by

certain elementary volume elements, like triangles-based, tesseroids, or hexahedra (*Forsberg*, 1984; *Werner and Scheeres*, 1996; *Nagy et al.*, 2000; *Heck and Seitz*, 2007; *Kuhn et al.*, 2009; *Grombein et al.*, 2014; *D'Urso*, 2014). The summation of individual volume elements multiplied with their density distribution is used to calculate a gravitational potential field. Any mass shape that can be approximated by the elementary bodies can be forward modelled into a gravitational potential. This technique is widely used, especially to model regional areas (*Forsberg*, 1984; *Kaban et al.*, 2010; *Holzrichter and Ebbing*,

2016). For global models, the computational time can become a complication, because higher resolution increases the amount of numerical integration rapidly (*Hirt and Kuhn*, 2014), unless multi-core computers are employed. The spectral forward modelling evaluates the Newton mass integral comparatively much faster by a transformation into the spherical harmonic domain (*Lachapelle*, 1976; *Rapp*, 1982; *Rummel et al.*, 1988; *Pavlis and Rapp*, 1990; *Novák and Grafarend*, 2006; *Root et al.*, 2016). The Fast Spectral Method (FSM) (*Rummel et al.*, 1988) has mostly been used for topographic/isostatic mass reductions

of the Earth to compute isostatic anomalies. *Root et al.* (2016) showed that the technique, with minor modifications, could be used to model any mass layer inside the Earth.

We present a benchmark study comparing three space-domain (triangles, tesseroids, hexahedra) and one spectral-domain approach applied to the layered WINTERC-G 3-D density model, in order to assess the usability of the model including the uncertainty resulting from different forward modelling approaches. While the focus of this benchmark are the different

parametrizations, we also need to address the approximations and inaccuracies of each individual method, to better appraise the differences between the methods. Therefore, we have carried out different tests, ranging from simple shell tests to a the more complex upper mantle model. We present the forward modelling scheme used in WINTERC-G and compare it to different forward modelling codes. Finally, the full 3D upper mantle model WINTERC-G is used as encompassing benchmark of the various forward modelling schemes in comparison to the XGM2016 gravity model that was used in the construction of

WINTERC-G (*Fullea et al.*, 2020).

## 2   Methods: Gravity integration techniques

The inversion code used to construct WINTERC-G relies on a spectral forward gravity modelling approach. The mathematical description of that method can be found in Appendix A. This choice of algorithm allowed for fast forward modelling, but tailored the density solution towards the spherical harmonic basis functions. To assess the applicability of this choice we select

four different forward modelling codes to understand the differences in forward modelled gravitational potential, resulting from WINTERC-G: the Global Spherical Harmonics spectral code based on *Root et al.* (2016), a tesseroid forward modelling code





based on *Uieda et al.* (2016), a triangle-element forward modelling code by *Sebera et al.* (2018), and a hexahedron-element code incorporated in geodynamical ASPECT modelling software (*Kronbichler et al.*, 2012; *Heister et al.*, 2017). This section will discuss the different forward gravity modelling schemes and their motivation to be included in this benchmark.

On the sphere, there is a wide range of point distributions that can represent the geometry of the shell (*Kimerling et al.*, 2008). The selected type of distribution is usually a trade-off between the number of points on a given domain (e.g., specific vs. homogeneous coverage), ease of manipulation (symmetries usually allow for a significant speed up) and the nature of data (e.g., nodal vs. volumetric). None of the spherical distributions is suitable for all the types of data that are used in geophysical/planetary sciences. For example, seismic models, that rather represent nodal information, may require a uniform

distribution on the sphere to provide the users with a minimal number of points in this domain - typically, triangular grids satisfy such needs. On the contrary, when density distribution models are to be used along with gravity, a common choice is to use grids that allow for accurate and fast volume integration - this especially holds for equi-angular grids as the surface elements can easily be translated into the volume elements (tesseroid). Note that the choice of the data representation on the sphere for different data types is still a subject of active research in fields like geophysics (*Thieulot*, 2018), astrophysics, computer

visualization, or mathematics.

## 2.1   Description of the GSH approach

The GSH code bench-marked here is based on the Fast Spectral Method described in *Root et al.* (2016). The code is written in MATLAB and is similar to the spectral method used in the development of WINTERC-G. The GSH code is capable of transforming a 3D-density layer with non-spherical boundaries into a gravitational potential signal. To process a multi-layered

density model (e.g., WINTERC-G) the derived Stokes' coefficients of the different layers are summed up before the total coefficients are used to synthesise the potential field of the entire density model.

The main difference between the GSH code and the spectral approach used in the development of WINTERC-G is the way the non-spherical boundary and lateral varying density are added. The GSH code adds these together before performing a spectral analysis on the combined function, whereas the WINTERC-based code performs the spectral analysis on the individual

components of the boundaries (Eq. (A12)). This different choice of performing the spectral analysis has consequences for the precision of the solution, as the spectral analysis is based on a least-squares fitting. The finite precision of this fitting process therefore results in slightly different solutions in the spectral codes. Nevertheless, the GSH code is expected to approach the WINTERC-G original code closest.

Another requirement of the GSH code is that the boundaries and density should be on an equi-angular grid, similar to the

WINTERC-G grid. *Root et al.* (2016) showed that the Fast Spectral Method had convergences issues for crustal and lithosphere layers and showed how to solve this problem. The GSH code corrects for these divergences, such that the solution remains accurate. For more information on this feature, see *Root et al.* (2016).





## 2.2 Integration by tesseroid elements

In the geoscientific community, the tesseroid algorithm of *Uieda et al.* (2016) has gained widespread popularity, as it is freely
available and has been thoroughly tested. The work of *Uieda et al.* (2016) is based on the adaptive subdivision algorithm of
*Asgharzadeh et al.* (2007), which guarantees a reasonable accuracy by splitting tesseroids into several smaller tesseroids based
on the distance between the calculation point and the tesseroid. Once the tesseroids are small enough, the gravity effects are
calculated using second order Gauss-Legendre Quadrature of the gravity integral and then summed up for all tesseroids.

A tesseroid is a spherical prism described by six values: its boundary coordinates in East, West, North, South, Top and
Bottom. Equi-angular tesseroids have a prismatic shape at low latitudes, but degenerate into increasingly triangular shapes
closer to the poles. In terms of Newton's integral (Eq. 1) the tesseroid approximation can be written as:

$$V(P) \approx G \sum_i \rho_i \tilde{K}(Q_i, P), \tag{2}$$

The index $i$ goes over all tesseroids used to discretize the model, $\rho_i$ is the constant density value of each tesseroid and $Q_i$
denotes its center. $\tilde{K}$ is a value of the kernel function derived from the adaptive Gauss-Legendre integration (*Grombein et al.*,
2013). While there is no closed analytical expression for the gravity field of a tesseroid (*Heck and Seitz*, 2007), a number of
techniques can be used to approximate it with a high accuracy (*Wild-Pfeiffer*, 2008). For large-scale applications, it can be
sufficient to use a truncated Taylor expansion of the integrand, but in practice this is limited to large calculation heights, since
higher-order expansions are cumbersome to determine and the expansions to second order are highly inaccurate close to the
tesseroid (*Heck and Seitz*, 2007).

## 2.3 Integration of the triangular grids

Numerical integration of the triangular grids in this work satisfies also Eq. (2) where the kernel function $\tilde{K}$ links the surface
(and thus volume) element of a triangular shape with a computational point. Here, the kernel is computed with respect to the
centre of mass of the spherical triangle, see *Sebera et al.* (2018). The density values are given for the nodes of the triangular
grid and not for the triangles themselves. This leads to a significant degree of freedom in the way the volume element is set
up around each node. The user has to decide where the element starts and ends with respect to surrounding triangles. The key
problem is that the spherical triangles do not have the same area. Hence, the surface and volume elements have different values
and the elements provide unequal weights to data in the numerical integration. To simplify the situation, here we assume the
triangular sides are great circles so that the spherical geometry can be employed for computing both the distance between the
nodes and the area of the element.

Figure 1 shows the situation for a single spherical triangle and its neighborhood. The red points define the grid (the given
density) of the WINTERC-G model (step 1) while the blue points denote a mid-point of each triangle. A centre of mass is
an ideal input if the density would be given for the whole triangle and not for the nodes in red. Although the area is uniquely
defined for each triangle, the triangle vertices may provide up to three different density values because of the lateral density
variation. Then, there are basically two ways to proceed before the integration, either to interpolate (average) the three density




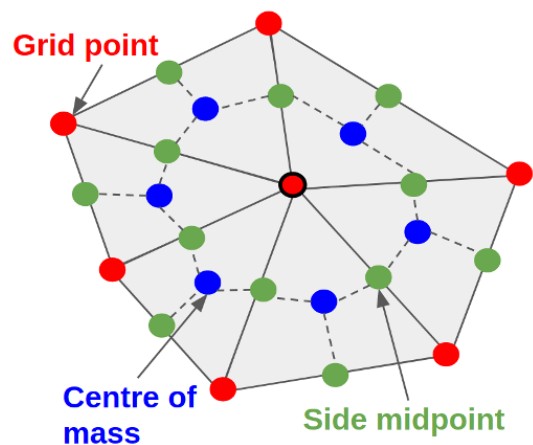

**Figure 1.** Sketch of the triangular grid on the sphere - grid/data points (red), triangle centre of mass (blue) and side midpoints (green).

values located at red points to obtain a single density for each triangle, or to build up an alternative area/volume element around the original nodes (around the red points). The first option leads to averaging of the physical information while the second requires to choose the geometry rules for setting up the surface/volume elements. In this work, we follow the second option to preserve the original density distribution from the benchmark model. The area associated with a nodal information is thus given as a local mean of the surrounding triangles according to Figure 1. However, this value must be scaled to the number

of points since the number of triangles is twice compared with the number of nodes in the global ($4\pi$) domain.

Figure 2 presents the triangular area variation for two different triangle grids on the sphere. The spline triangular grid used in WINTERC-G (*Wang and Dahlen*, 1995) is compared to a icosahedron grid *Pasyanos et al.* (2014). For both we can see a variation of up to 20% in the area, whereas the grids significantly differ in the area gradient (smooth vs. sharp variation). The triangle grid used in WINTERC-G shows more smooth transitions, than the isosahedron grid.

In the first step WINTERC-G development uses a triangular grid to invert seismic tomography, surface heat flow and isostasy data whereas in the second step the gravity field is modelled based on a spherical equi-angular grid to accommodate the fast spectral code. The specific challenge is the calculation of gravity from a triangular grid because there is no triangular grid that would be perfectly uniform on the sphere. The nodes associated with larger triangles thus produce a larger signal so that the results are then systematically affected by the triangular patterns.

## 2.4 Description of the gravity post-processor in the ASPECT code

ASPECT (short for Advanced Solver for Problems in Earth's ConvecTion) is a code originally intended to solve the equations of conservation of mass, momentum and energy in the context of convection in the Earth mantle and lithosphere dynamics (*Kronbichler et al.*, 2012; *Heister et al.*, 2017). It is a massively parallel Finite Element (FE) code which relies on the `p4est` library (*Burstedde et al.* , 2011) for the handling of the Adaptive Mesh Refinement based on quad/octrees. As part of any Finite

Element code the Gauss-Legendre Quadrature (GLQ) algorithm is central to the code.





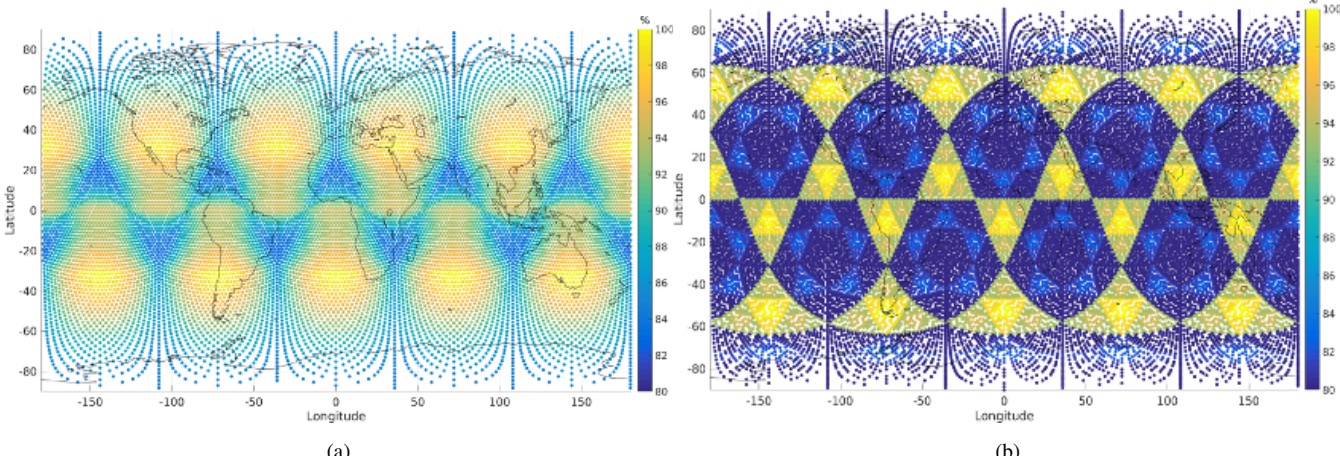

**Figure 2.** (a) Triangles area for the spline triangular grid (*Wang and Dahlen*, 1995) and (b) the icosahedron grid (*Pasyanos et al.*, 2014). Both grids were evaluated in terms of geodesics (triangle sides are treated as great circles on the sphere) and both panels shows the area variation in percent with respect to the largest triangle.

Given a density field in the computational mesh, ASPECT can also compute the gravity acceleration vector, the gravitational potential and the gravity gradients on any point in space. Since the integrand of the integral equations is not a polynomial the GLQ-based computed integral will not be exact. Nevertheles, we expect that an increase in the number of quadrature points inside the elements leads to a more accurate calculation. ASPECT relies on quadratic elements for velocity and temperature, an array of $3 \times 3 \times 3$ quadrature points is used in each element by default. After careful testing, we have chosen to use a $6 \times 6 \times 6$ quadrature rule in each element, since an increase did not yield a substantial change in the results. Each integral equation is approximated by Eq. (3):

$$I(\boldsymbol{r}) = \iiint\limits_{\Omega} f(\boldsymbol{r}, \boldsymbol{r}', \rho(\boldsymbol{r}')) d\boldsymbol{r}' \simeq \sum_e \sum_q f(\boldsymbol{r}, \boldsymbol{r}'_q, \rho(\boldsymbol{r}'_q)) |J_e|_q \omega_q \qquad (3)$$

where the first summation runs over the elements $e$ and the second summation runs over the quadrature points $q$ inside element $e$, $\omega_q$ is the weight of the quadrature point, $|J_e|_q$ is the Jacobian of the mapping of the element onto the reference element and $f$ is a function of space (the integrand).

The (default) topology of the mesh in ASPECT is shown in Figure 3, which is based on a decomposition of the sphere into six identical regions as described in (*Thieulot*, 2018). In the case of the shell tests described in the following section, a single element is used in the radial direction. The total number of elements in the mesh is then simply $n_{el} = 6 \times (2^m)^2$ where $m$ is the lateral refinement parameter, with $m = 6$ being default. Figure 3 also shows the variable radial layering parameterisation for the full WINTERC-G benchmark. An increased amount of layers is chosen up to 80 km depths, to have more radial resolution to capture the lateral varying density interfaces of the crust.




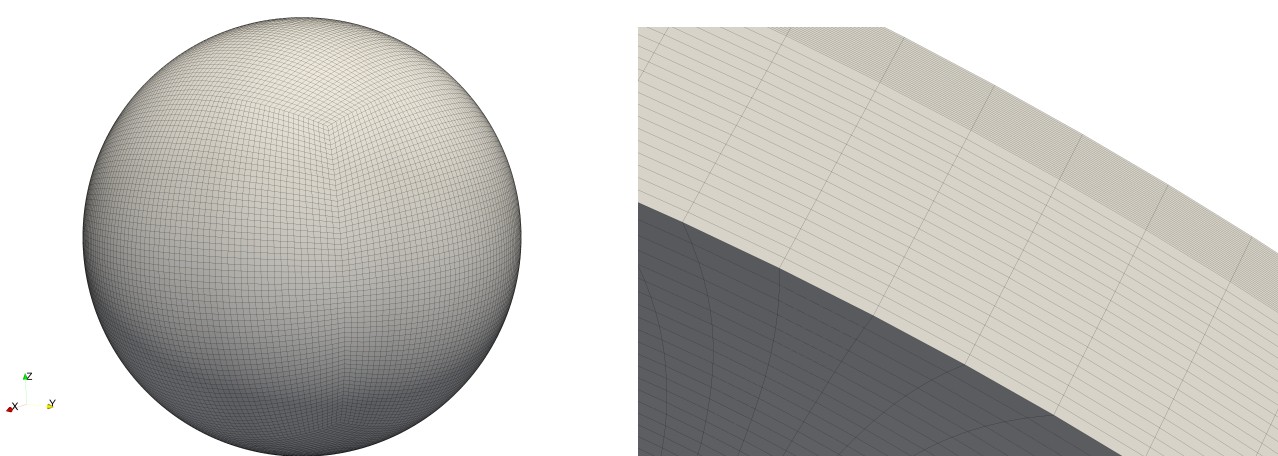

**Figure 3.** Left: Topology of the ASPECT mesh composed of 6 blocks. Right: cross section of the mesh with radial refinement as used in Section 4 (here with 30 elements in the top 80 km and 20 elements below 80 km depth).

## 3   Preliminary single shell comparisons

The most basic shell test is a spherical shell with finite thickness and a constant density. Here, we mainly assess the volumetric-based approaches and their resolution, because spectral-type codes have an exact solution for a homogeneous density shell down to machine precision. Therefore, two other shell tests have been proposed : an equal thickness shell with laterally varying density and a shell with a depth-varying density discontinuity. The WINTERC-G model is described by layers with laterally varying density as well as laterally varying density interfaces (e.g., surface topography, basement, Moho discontinuity) .

### 3.1   Shell test 1: Equal thickness and homogeneous density

The gravity field of a homogeneous spherical shell can easily be calculated analytically, because due to symmetry the relationship only depends on the radial distance of the computation point:

$$g(r) = \frac{4}{3}\pi G \rho \frac{R_2^3 - R_1^3}{r^2}, \qquad (r \geq R_2), \tag{4}$$

where $G$ stands for the gravitational constant, $\rho$ for the density of the shell, $R_1$ and $R_2$ for the inner and outer radii of the shell respectively and $r$ for point of evaluation. This simple geometry provides a suitable means for testing the performance of the different integration schemes. We place spherical shell at a mean depth of $100\,\mathrm{km}$ with respect to the Earth's $6371\,\mathrm{km}$ reference sphere and modelling different thicknesses of 2, 5, and 10 km. The density of the shell is equal to $3300\,\mathrm{kg\,m^{-3}}$ and the altitude of calculation is $250\,\mathrm{km}$ above the $6371\,\mathrm{km}$ reference surface. The results are summarized in Table 1 rounded to 3 digits precision. We observe that all numerical schemes can provide gravity values that are very close to those of the analytical solution.





As expected, the GSH code produces a solution similar to the analytical value within machine precision, roughly a standard deviation of $10^{-9}$ mGal, independently of the thickness of the layer. The spherical harmonic analysis applied to a shell of constant density returns an exact value of the density at the mono pole coefficient (single frequency).

For the tesseroids, a small bias was found at the fifth significant digit indicating a very good overall performance across all the latitudes. There is only a negligible variation close to the poles as seen in Figure 4. The solution deteriorates due to the

degenerated tesseroid shape approaching the poles. However, such error is not expected to cause problems in the integration of real global density distribution models since the uncertainty of such models is orders of magnitude larger than this numerical issue. For example the typical posterior uncertainties in crustal density in WINTERC-G range from 5 to $25 \, \mathrm{kg\,m^{-3}}$ *Fullea et al.* (2020). The tesseroid code calculates signals with sub-mGal differences with respect to the analytical solution. The increasing thickness of the shell seems to improve the precision slightly. There is a slight under-determination of the analytical

signal, indicating a small volume loss in the tesseroid code. This is related to the limitations of the numerical integration of that approach.

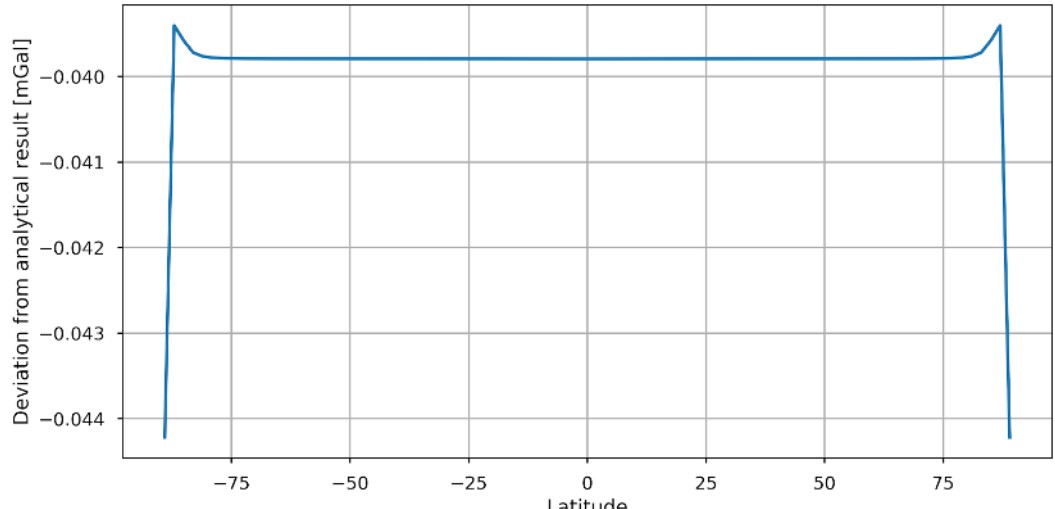

**Figure 4.** Deviation of the tesseroid integration from analytical shell result as a function of latitude. This is for the case of the 2 km thick shell with a 1 degree lateral resolution.

The triangle-based approach seems to perform slightly better than the tesseroid approach, but does show different behavior depending on the lateral resolution of the triangle grid. At roughly 2° grid, the RMS value is close to the analytical signal, but the solutions experience large variations. The 1° grid does reduce this variation slightly, with the cost of larger differences in

the RMS value. The 0.25° grid reduces the variation and differences in the RMS and obtains a precision at several µGal. The triangular elements differ in volume and, because the kernel associated with the volume elements is related to a single point (the midpoint of the volume element), some irregularities are expected.





| Thickness (km) | 2 | 5 | 10 |
|---|---|---|---|
| Shell formula (mGal) | 496.574771345 | 1241.43698361 | 2482.87436182 |
| GSH (1°) | 496.574771345, $\sigma = 10^{-10}$ | 1241.43698361, $\sigma = 10^{-9}$ | 2482.87436182, $\sigma = 10^{-9}$ |
| Tesseroid (1°) | 496.540 ±0.005 | 1241.348 ±0.006 | 2482.696 ±0.007 |
| Triangles ($\sim 2°$, opt. 2) | 496.576 ±0.2 | 1241.441 ±0.5 | 2482.883 ±1.0 |
| Triangles ($\sim 1°$, opt. 2) | 496.535±0.05 | 1241.334 ±0.11 | 2482.674 ±0.23 |
| Triangles ($\sim 0.25°$, opt. 2) | 496.572 ±0.003 | 1241.431 ±0.007 | 2482.862 ±0.014 |
| ASPECT ($m = 4, \sim 5.6°$) | 496.575 ±0.0243 | 1241.437 ±0.068 | 2482.875 ±0.12 |
| ASPECT ($m = 5, \sim 2.8°$) | 496.575 ±3.5 × $10^{-5}$ | 1241.437 ±8.8 × $10^{-5}$ | 2482.874 ±1.8 × $10^{-4}$ |
| ASPECT ($m = 6, \sim 1.4°$) | 496.575 ±1.0 × $10^{-7}$ | 1241.437 ±1.0 × $10^{-7}$ | 2482.874 ±1.0 × $10^{-7}$ |

**Table 1.** Summary of the homogeneous density shell test at the target height $h = 6371 + 250 = 6621$ km with $G = 6.67428 \cdot 10^{-11}$ $\mathrm{m^3\,kg^{-1}\,s^{-2}}$. The numbers indicate average residual and their maximum variation. Note that ASPECT uses by default $G = 6.67430 \cdot 10^{-11}$ $\mathrm{m^3\,kg^{-1}\,s^{-2}}$ so that the values reported in the table are re-scaled for the chosen value of G.

The ASPECT code outperforms the triangle and tesseroid approach concerning precision and lateral resolution. It achieves µGal precision at 5° resolution and the increase of resolution improves the solution with respect to the analytical case. For higher resolution grid the solution of the ASPECT code approaches the analytical solution, with 1.4° having residuals of less than $10^{-7}$ mGal.

So, all four codes are able to obtain the gravity signal up to µGal precision, concluding that the volume losses can be neglected in between the four forward modeling schemes for density models with typical geophysical uncertainties.

## 3.2 Shell test 2: Equal thickness and lateral varying density

The following shell test examines the capability of the different forward modelling schemes to handle lateral density variations. We will first show that the GSH code is capable of producing similar results compared to the WINTERC-G based code. Then, we discuss the difference between the four modelling approaches in this benchmark.

The mass shell in this scenario consists is described by the outer radius, located at 56 km depth, and the inner radius, located at 80 km depth. The density values within the layer correspond to typical lithospheric scale lateral density variations in WINTERC-G model, as presented in Figure 5a. The densities range between 3286 and 3419 $\mathrm{kg\,m^{-3}}$ and still show some correlation to the geological crustal structures above. In the comparison with the WINTERC-G based code, we compute the geoid anomaly for a cut-off degree of Stokes potential coefficients ($j_{max}$) at 240 degree and order. The geoid is defined as:

$$N(\Omega) = R \sum_{j=1}^{j_{max}} \sum_{m=-j}^{j} \left(\frac{R}{r}\right)^{j=1} V_{jm}^{\rho} Y_{jm}(\Omega), \qquad (5)$$

Notice that the summation starts at $j = 1$ to obtain the geoid undulation.

The forward modelled geoid differences of this layer between the WINTERC-G code and GSH code are shown in Figure 5b. The total geoid undulations vary between ±300 m. The peak-to-peak residuals are maximum ±2 cm geoid differences, but with a standard deviation of around ±1 cm. This difference is generated by the variation of the spherical harmonic analysis that





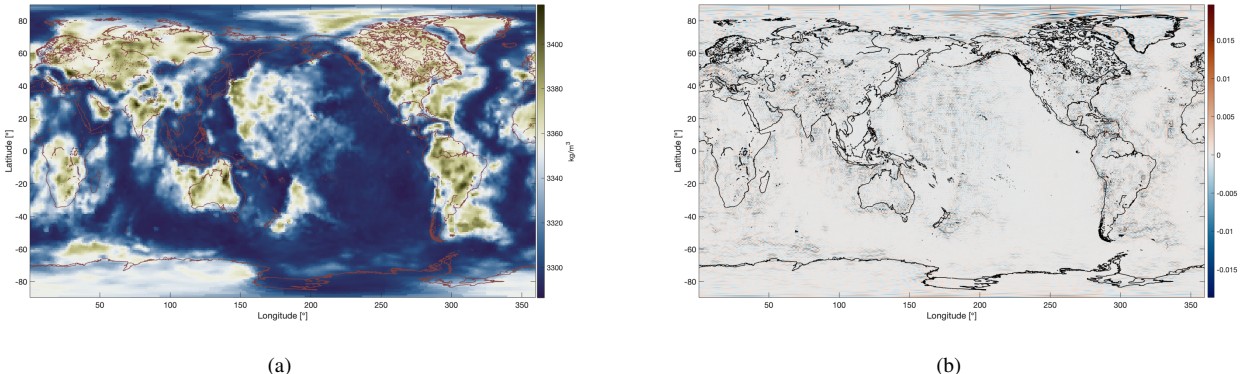

(a)                   (b)

**Figure 5.** (a) WINTERC-G density variations in the mantle from the WINTERC-G model. These were used in the spectral shell comparison as radially-constant density variations between 56 and 80 km depths. (b) Geoid differences of the lateral varying density from Figure 5a between the WINTERC-G based code and the GSH code.

produces the spherical harmonic coefficients in both codes. This procedure is not exact and differs slightly due to numerical precision (*Sneeuw*, 1994) and when applied at a different part of the integration leads to small errors, especially at locations

250 where the density is varying. Nevertheless, the spectral forward modelling schemes are able to represent potential fields that are well within the geophysical uncertainties. And the GSH approach can be considered to be equal to the WINTERC-G based approach.

This same shell is processed with the other forward modelling approaches. The radial component of the gravity field is computed at $250 \, \mathrm{km}$ height above the mean sphere, as this was the height on which the satellite gravity data for the development

of WINTERC-G was used. Shell test 1 showed that the mean gravity uncertainty between the different numerical codes, which is linked to the zero-degree coefficients, was insignificant. Therefore, the spherical harmonic coefficients 2-179 were used to focus more on the anomalies of the gravity solution. This meant that the solutions had to be post-processed by the GSH code to ensure that a similar spectral signature is used in the comparison. This introduced some errors at machine precision level.

Figure 6 visualises the differences of various the forward modeling results. The total radial gravity anomalies of this shell

model vary between $\pm 50 \, \mathrm{mGal}$ and the spatial pattern matches the density pattern in Figure 5a. Continental (cratonic) regions are characterized by cooler and denser rocks than oceans in general, resulting in positive gravity anomalies. In contrast, oceanic regions are associated with negative anomalies. The differences between all four codes mostly fall between $\pm 1 \, \mathrm{mGal}$, which is around 2% of the total gravity signal. These differences are larger than the difference between the two spectral codes. The lowest residuals are found between GSH and TESS where a residual of $\pm 0.6 \, \mathrm{mGal}$ (around 1%) with no apparent geologic

pattern is observed. The north-south oriented pattern present in the residual anomalies are mostly situated in the equator region and point to a small mismatch in the interpolation of the density structures between GSH and TESS codes. Larger residuals are seen in the comparison with respect to the other codes. The most prominent erroneous features are the triangle-related





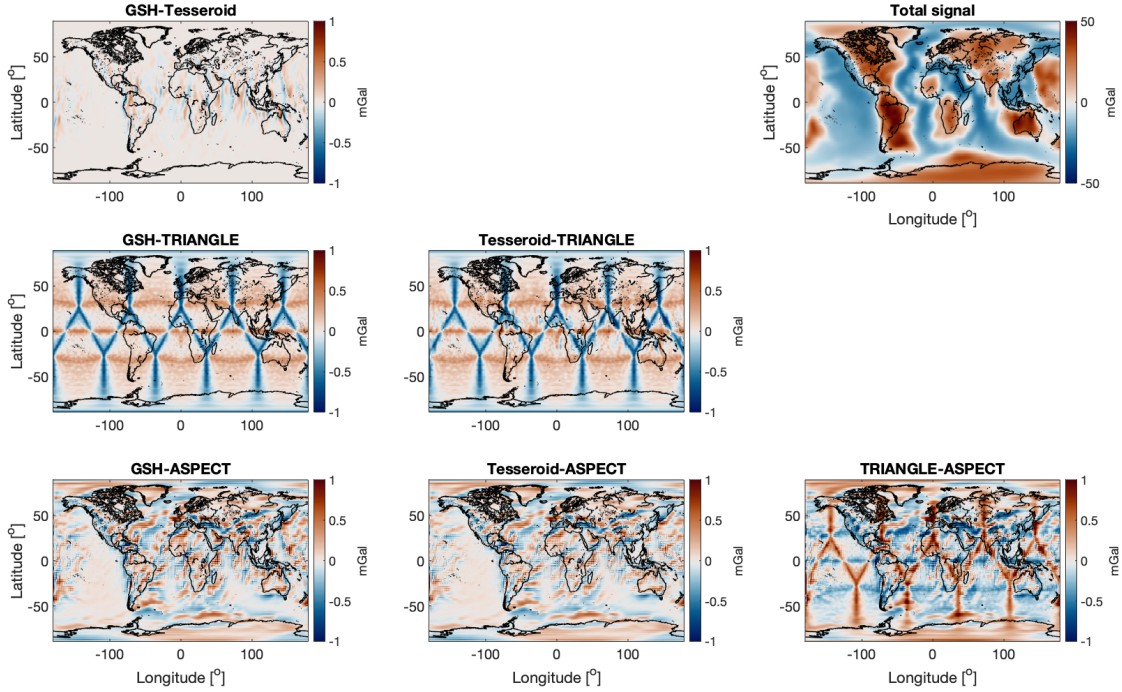

**Figure 6.** Radial gravity component comparisons at 250 km height for the shell test 2, where a 24 km thick shell is modelled with lateral varying density structure. A grid resolution of 1x1 equi-arc degree was used for GSH, TESS, and TRI. ASPECT was run on a L7 mesh, i.e. an average lateral resolution of less than a degree.

pattern of $\pm 1.0$ mGal in comparisons with TRI and any other code. Such global patterns are related to imperfect volume representation by the triangles of the spherical shell in the triangulation already discussed in Section 2.3. The characteristic

north-south oriented anomalies are also seen around the equator region in the TESS-TRI comparison (but not in the GSH-TRI comparison), suggesting that these anomalies are generated by TESS density interpolation. The ASPECT code produces slightly smaller differences than TRI. The difference between ASPECT and the GSH and TESS approaches seem to have some correlation with the input density model although this is not everywhere obvious. A pattern that correlates with the Mid-Atlantic ridge in the north Atlantic ocean, suggest improper modelling of this geologic structure by the ASPECT approach. However,

there are also areas (e.g. East Pacific) where no apparent correlation with the density distribution is seen. Furthermore, the residuals seem to be more east-west oriented than in the case of the TESS solution.

     Table 2 lists the statistical analysis of the gravity solution differences of shell test 2. The absolute variations in residuals with the ASPECT solution contain the largest outliers, despite the fact that the standard deviation of the ASPECT comparisons are slightly better than for the TRI approach. However, they are small and both ASPECT and TRI could be said to perform at

similar precision ($\pm 0.2$ mGal standard deviation). The superior performance of the GSH and TESS approaches is noticeable in





the standard deviation ($\pm0.055$ mGal) as well as the minimum and maximum differences between both solutions ($< 1$ mGal).

| Solution (mGal) | Mean | Std. | Min | Max |
|---|---|---|---|---|
| Total signal | 1.4921 | 17.7815 | -31.9188 | 53.3946 |
| GSH - Tesseroid | 0 | 0.055214 | -0.46545 | 0.82247 |
| GSH - Triangle | -0.02539 | 0.21548 | -0.97375 | 0.55155 |
| GSH - ASPECT | 0.001365 | 0.21479 | -1.6873 | 3.7413 |
| Tesseroid - Triangle | -0.02540 | 0.22492 | -1.4898 | 0.81721 |
| Tesseroid - ASPECT | 0.001365 | 0.19245 | -1.3774 | 2.9231 |
| Triangle - ASPECT | 0.026764 | 0.30615 | -1.8461 | 4.3829 |

**Table 2.** Statistical results from the shell test 2: a density shell of equal thickness is modelled with lateral varying density structure.

### 3.3 Shell test 3: Density contrast at the Moho

High-resolution lithosphere/upper mantle models combined with increasing computing capabilities offer new possibilities in
relation to dynamic studies like mantle convection, GIA, or geo-hazards. Geometric boundaries within the crust and mantle
that vary in depth and thickness are difficult to represent in numerical models. Nevertheless, such discontinuities produce large
gravitational signals and, hence density boundaries need to be represented as perfectly as possible as slight changes to their
depths could have noticeable effects in the full lithospheric gravitational signal. In particular, the top and bottom boundaries
of the crystalline crust (basement and Moho boundaries respectively) are of importance. In this test, we model a single density
interface, representing the crust-mantle interface taken from the CRUST1.0 crustal model (*Laske et al.*, 2013). The shell
thickness is 80 km. The crustal density (2900 $\mathrm{kg\,m^{-3}}$) and mantle density (3300 $\mathrm{kg\,m^{-3}}$) are homogeneous, resulting in a
gravitational signal coming solely from the interface geometry.

   Shell test 3 assesses the precision of the different codes in modelling a geometrically-varying density interface. The results
are depicted in Figure 7. The total signal of the shell model is between $\pm400$ mGal representing the gravitational signal as
a result of the Moho density contrast. The continental regions are characterised by negative gravity anomalies because of the
thicker crustal mass with lower density. Most of the oceanic regions are showing positive gravity anomalies in virtue of their
shallow Moho depth. The GSH-TESS differences are again smallest with values ranging between $\pm0.1$ mGal. The largest
differences seem to be related to the locations where the Moho boundary changes much (e.g. ocean-continental interfaces).
This observed difference can be related to the fact that TESS and GSH methods approximate the boundary with different basis
functions and are highest at locations with steep boundary variations. The residual signal calculated with TRI code shows a
similar characteristic structure as in shell test 2. A clear triangle-related pattern is present as shown in Figure 6. The largest
observed amplitude in the residual of $-4.3$ mGal is in this case 1 percent of the total signal. This is relatively smaller than
in the shell test 2 ($< 2\%$) with the lateral varying densities, this smaller effect is due to the larger shell thickness of test 3
(80 km $>$ 24 km). The GSH and TESS methods are more compatible representing laterally varying density discontinuities




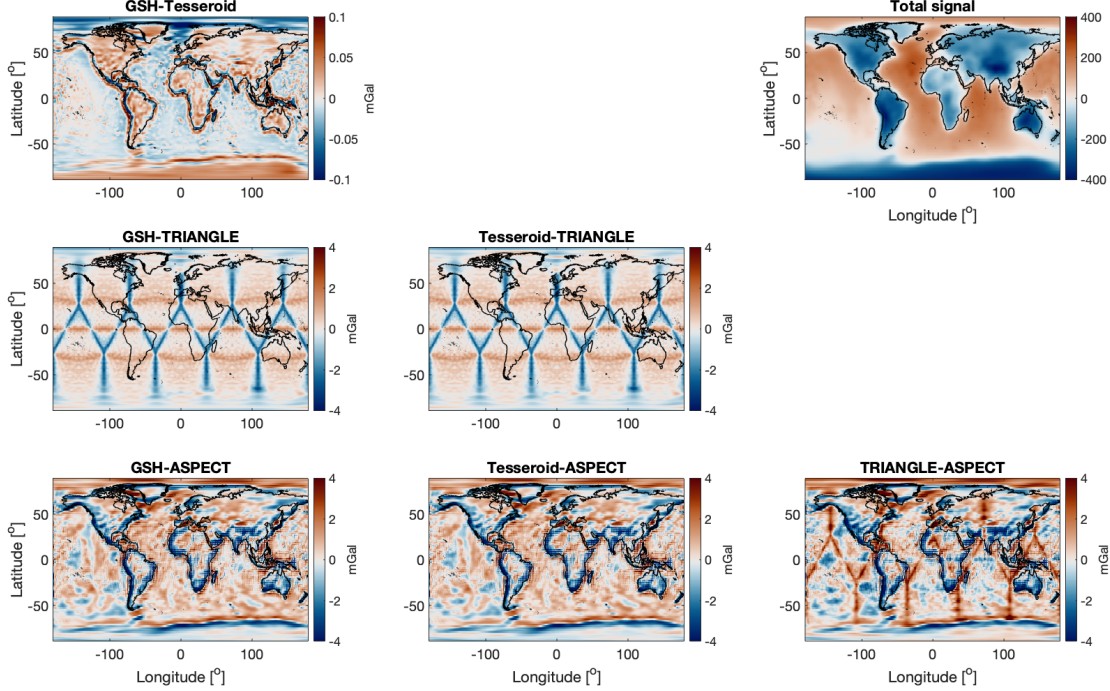

**Figure 7.** Radial gravity component comparisons at 250 km height for the shell test 3, where a density shell of equal thickness is modelled with a density contrast at the CRUST1.0 Moho boundary.

than lateral varying density fields. The difference between ASPECT and the other codes differ roughly about $\pm 4$ mGal, but have some larger outliers going up to $-15.5$ mGal (listed in Table 3). The differences using ASPECT code show some more correlation with the input density/geometry structure, for example in the Pacific ocean correlations to the tectonic plate boundaries are seen in the residuals. Because the ASPECT code performed better in shell test 2, these larger residuals suggest that the ASPECT results tend to be limited by representation of the geometry of the density interface. The chosen lateral

resolution (L7) results in variations similar to the triangle-based code, so it can be pin point to the radial resolution. Because the ASPECT code can only incorporate equal thickness layers, it needs to represent the Moho geometry with different equal thickness layers. For this particular result the 80 km thick shell was divided into 40 layers, which results in a radial resolution of 2 km. This is further discussed in Section 5.

    The results in Table 3 agree with the observations from Figure 7 and highlights the outliers in the ASPECT code comparison.

Overall, the standard deviation of all comparisons fall well below the 1 percent of the actual signal to model and the mean signals are quite consistent with each other. The ASPECT code seems to be the least capable to forward model density contrasts of boundaries with varying depths, which related to the limited radial resolution. Together with the observed performance of lateral varying densities, we expect that the ASPECT code will be least able to predict the main gravity field features associated





| Solution (mGal) | Mean | Std. | Min | Max |
|---|---|---|---|---|
| Total signal | -27.7221 | 150.7689 | -408.1635 | 261.9314 |
| GSH - Tesseroid | 0 | 0.026893 | -0.13836 | 0.16555 |
| GSH - Triangle | -0.072622 | 0.75468 | -4.3124 | 1.9582 |
| GSH - ASPECT | 0.0053177 | 1.4619 | -14.1503 | 11.547 |
| Tesseroid - Triangle | -0.072622 | 0.75392 | -4.3593 | 1.9709 |
| Tesseroid - ASPECT | 0.0053177 | 1.4546 | -14.0777 | 11.4364 |
| Triangle - ASPECT | 0.07794 | 1.688 | -15.5443 | 11.3065 |

**Table 3.** Statistical results from the shell test 3: a density shell of equal thickness is modelled with a density contrast at the CRUST1.0 Moho boundary.

with WINTERC-G lithospheric/upper mantle density model because of this. Nevertheless, the differences between each code
under benchmark here can be considered a reliable assessment of the general applicability of variable geometries in lithosphere
models using different techniques.

## 4  Whole WINTERC-G density model integration

The choice of forward modelling scheme and parametrization of similar density models could lead to non-negligible local differences in the modelled gravitational signal. This could, if not properly understood, lead to erroneous interpretation of
geological structures. With increasingly high resolution gravity data sets and their associated density models this becomes an important technical modelling issue. In this section, we study the forward gravity modeling of the whole WINTERC-G model, the data set can be found at *Fullea et al.* (2021). We will solely focus on the density and geometry assets of this model, as they are related to the gravitational potential field.

WINTERC-G model goes through different vertical and horizontal parametrisations during its two-stage inversion process
(*Fullea et al.*, 2020). In the first inversion stage, the model consists of a collection of 1-D columns distributed on the sphere as a triangular grid according to *Wang and Dahlen* (1995), also used in the phase velocity maps of *Schaeffer and Lebedev* (2013). Vertically, the model has a hybrid parametrisation that combines a 2 km step regular sampling with variable depths for the surface, Moho and lithosphere-asthenosphere boundary (LAB) depth. This hybrid approach is necessary because the finite differences thermal solver uses a regular vertical grid, but most of the the modelled data sets (seismic, gravity, isostasy) are
very sensitive to the depth of physical discontinuities that must, therefore, be parameterised separately. In the second inversion stage, the density structure at each column is simplified to accelerate the gravity calculation. A series of layers are defined by spatially variable tops and bottoms and the average density in that depth range calculated from the 2 km spacing interval or a linear gradient between the top and bottom values is used. Except for the water-rock interface, the ice-rock interface and the Moho boundary depth, these layers are spherical shells with equal thickness. The Moho depth is highly variable, so the crustal





layers may cut into the shells, leading to a more complicated vertical structure in the upper 80 km. The depth and density values
from the columns are then interpolated onto an equi-angular grid and used for gravity inversion.

Hence, from a gravitational point of view, WINTERC-G density model consists of 13 layers (see Table C1), which are
defined by top and bottom boundaries and density distributions. The first 7 layers of WINTERC-G have varying thicknesses
caused by the varying geometries of these boundaries with respect to constant radius surface. These layers describe the structure

of the model from the top of the topography to the first 80 km depth (deepest Moho variations) and contain the water and ice
layer on top of a crustal layer. The other layers cut the sub-Moho region up to 80 km into 4 layers to have an increased radial
resolution as this region was found to be of importance to the gravity field (*Fullea et al.*, 2020). The other 6 layers are computed
using a constant radius for the boundaries, making layers of equal thickness. These mantle layers go up to a depth of 400 km
comprising the whole upper mantle. A detailed description of all layer discontinuities and density distributions can be found in

*Fullea et al.* (2020).

### 4.1 WINTERC-G model-based gravity signal

Here, we compute WINTERC-G associated gravity signal by means of an independent gravity approach in order to asses the
reproducibility.The calculated signal should match the gravity data inverted to build WINTERC-G, that is XGM2016. We use
the GSH software for this, as it resembles the most the spectral code used for WINTERC-G.

The GSH software produces a geoid solution by computing the potential field from WINTERC-G and then divides this by
$9.81 \, \mathrm{m \, s^{-2}}$. The normal gravity field needs to be removed from the observations by subtracting the fully-normalized coefficients
from the GRS80 ellipsoid (*Moritz*, 1980): $C_{00}$ is 1.00, $C_{20}$ is -4.842e-04, $C_{40}$ is 7.903e-07, $C_{60}$ is -1.687e-09, and $C_{80}$ is
3.461e-12. Furthermore, WINTERC-G is not using coefficients up to degree 4, so degrees 0-3 are discarded in this comparison.
The WINTERC-G-GSH comparison needs to take the divergence criterion into account (*Root et al.*, 2016). The crustal layers of

the WINTERC-G model are therefore cut in 4 internal layers for the GSH scheme. With this modification, the GSH code is able
to produce similar gravity potential to XGM2016 observations as the WINTERC-G geoid by its own internal code, as shown
in Figure 8. The total signal of the WINTERC-G model gives $\pm 100 \, \mathrm{m}$ geoid undulations, which resemble the observed geoid.
The difference between XGM2016 and the WINTERC-G are mainly long-wavelength variations of around $\pm 2 \, \mathrm{m}$ are similar
to the residuals reported in *Fullea et al.* (2020). Larger magnitude differences around $\pm 8 \, \mathrm{m}$ are small scale features, most

prominent in the Polar, Himalayas and Andes regions. The difference between XGM2016 and the GSH result show that the
GSH solution is slightly better approaching the XGM2016 observations than the WINTERC-G results. The residual between
WINTERC-G and GSH is below the misfit of XGM2016 with WINTERC-G and only show small-wavelength differences
(approx. 359 SH degree and order). Overall, this comparison shows that the GSH code is able to represent the WINTERC-G
model within similar precision as the code used by *Fullea et al.* (2020) in the development of WINTERC-G.

### 4.2 Forward modelling and comparison of the WINTERC-G gravity fields

In order for WINTERC-G to be useful for independent gravity-based research, the gravity field computed by our different
forward modelling approaches (covering most of the commonly used techniques in solid Earth modelling) should be below the





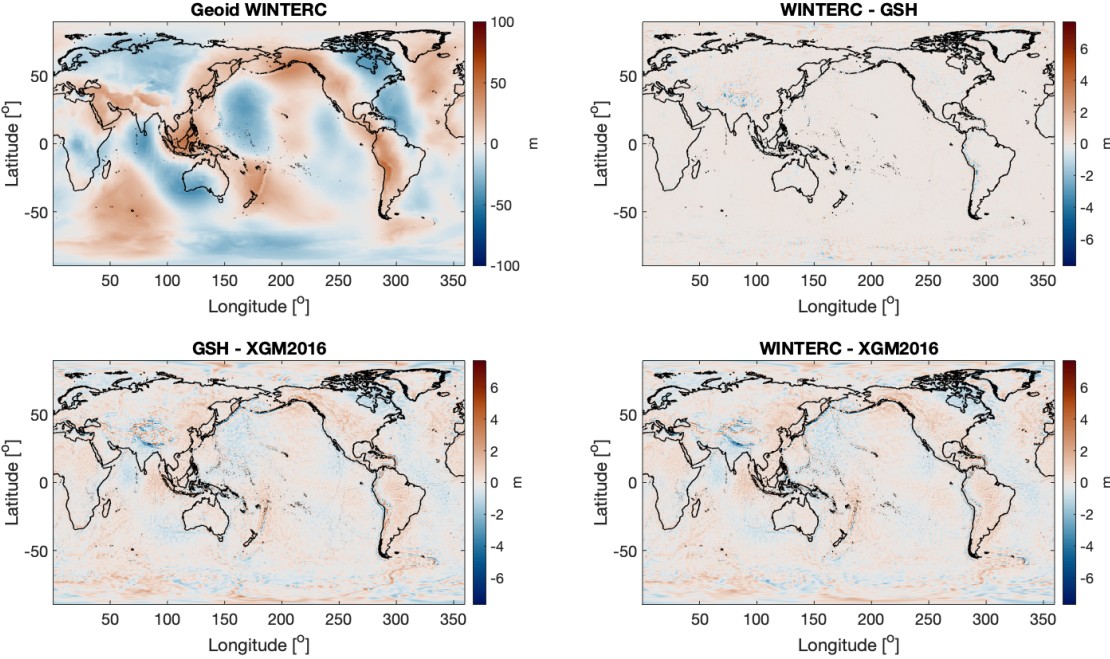

**Figure 8.** Geoidal differences comparing the solutions made by the WINTERC-G-based code, the solution from GSH, and the observed XGM2016 gravity field on which WINTERC-G is based. Spherical harmonic coefficients 4-359 degree and order are shown.

differences between WINTERC-G and XGM2016. In this section, the full WINTERC-G model is forward modelled into the gravitational field by the selected methodologies. The approaches were kept free in selecting best parameters (e.q. resolution, meshing) for the forward modelling result. The resulting radial gravity vector component would be examined at 250 km height above the reference sphere of 6371 km radius and only SH degrees 2 to 179 were taken into account. The reduced spectral resolution of 179 degree instead of 359 degree was chosen because of reduction in computation time. The signal above 179 degree has limited strength at 250 km altitude. This would result in differences of WINTERC-G and XGM2016 of a standard deviation of around 2 mGal, so all codes should be below these values.

Figure 9 shows the radial gravity vector component of the forward-modelled WINTERC-G density model (GSH result is depicted). The model produces gravity anomalies of $\pm 50$ mGal corresponding to the anomalies of the observed gravity field (XGM2016). The difference between the GSH and TESS solutions are $\pm 0.3$ mGal. These differences are below 1% of the actual signal, as could be expected from the simple shell tests. Furthermore, these differences are well below the data model uncertainty ($\pm 2$mGal) of WINTERC-G with respect to XGM2016. The residual signal reflects theeffect of the volume approximation of the tesseroid and spherical harmonics for the boundary geometries. The difference are smaller than in the tesseroid-GSH code benchmark in *Root et al.* (2016) using an earlier version of the GSH approach (there the differences were








around 10%) showing that the current GSH approach has improved. The differences between the GSH and tesseroid codes are largest in the continental regions and show some correlation with crustal structures that have large density gradient values. Especially, the active Pacific American margin, as well as the Himalaya region, experience noticeable residual anomalies. This
feature was seen in Shell test 3 and was attributed to dissimilar approximations of the geometrical boundaries.

    The triangle integration, similarly to the results from previous tests, yields a radial gravity component very close to those from GSH and tesseroid codes: no apparent differences are visible except for the characteristic triangle features also seen in the shell tests. The residuals with GSH and tesseroids show that the triangle integration probably got close to the limit of this technique because we can see triangular artifacts only (see the mid-latitudes in Figure 9). These artifacts were present in all
previous results and comes from the fact that the global triangular grid is not perfectly regular on the sphere (see Figure 2). To some extent, the triangular effects can be removed with the spectral filtering but such filter only helps with short-wavelengths effects (*Sebera et al.*, 2018). The triangle technique is surprisingly accurate given the fact that the integration kernels are calculated point-wise (one point value for each volume element). The triangular integration even helps identifying small crustal-correlated effects in GSH solution, notice the anomalies around the Himalayan region in the GSH-TRI comparison which
are not present in the TRI-TESS comparison. Nevertheless, due to the volumetric differences the triangle integration was performed with a standard deviation of $< 1$ mGal that is about 10 times larger than the standard deviation between GSH and TESS, see Table 4. For the triangles a spatial resolution of 0.5 arc-deg degree was used (the so-called Level 8 equipped with 196,002 nodes). Going even higher would dramatically increase computation costs and limit the performance of useful forward modelling.

This is also the case for the ASPECT code, which has the largest differences (0.8 to 1 mGal standard deviation, with outliers up to 10 mGal). Unless a high resolution is used, the ASPECT code has difficulties in obtaining the correct gravity signal, mostly related to representing the various boundaries: surface, Moho, ice-bedrock and other boundaries. This is mainly attributed to a lack of adequate radial resolution, as explained in Section 3.3. We have here made use of a code feature which allows the user to prescribe the radii of the concentric layers of nodes making the mesh. The lateral resolution is still level
6 (i.e. $6 \cdot 64^2$ elements per shell - approximate $1.4°$ resolution) but a higher radial resolution is prescribed in the first $80$ km. The results shown in Figure 9 were obtained with 90 layers in the top $80$ km (sub-kilometer resolution) and as many below it ($\sim 3.5$ km resolution). A python code was written to convert/re-sample the WINTERC-G data in a format readable by ASPECT (see Supplementary material). In the end the mesh counts about 4.4 million elements. Several dozens of Gb of RAM are then needed to run the code.

Table 4 depicts a numerical summary of the results of the WINTERC-G benchmark for all the different codes. The GSH and tesseroid codes produce similar results, although there are some outliers of $\sim 2.3$ mGal, which are mainly situated in the region of the Himalayan Mountains. Here, the crustal structures experiences the largest gradients in geometry, where we would logically expect the largest differences between the two codes. The other codes show larger deviations, with the GSH-triangles having an standard deviation of $0.7$ mGal, ten times larger than with the GSH-tesseroid comparison. However, the outliers
are similar, approximate $\pm 2.5$ mGal. The errors due to the difference in spherical harmonics representation of the boundaries is similar to the error made by differences in tesseroid or triangle choice. This shows that globally all three codes produce



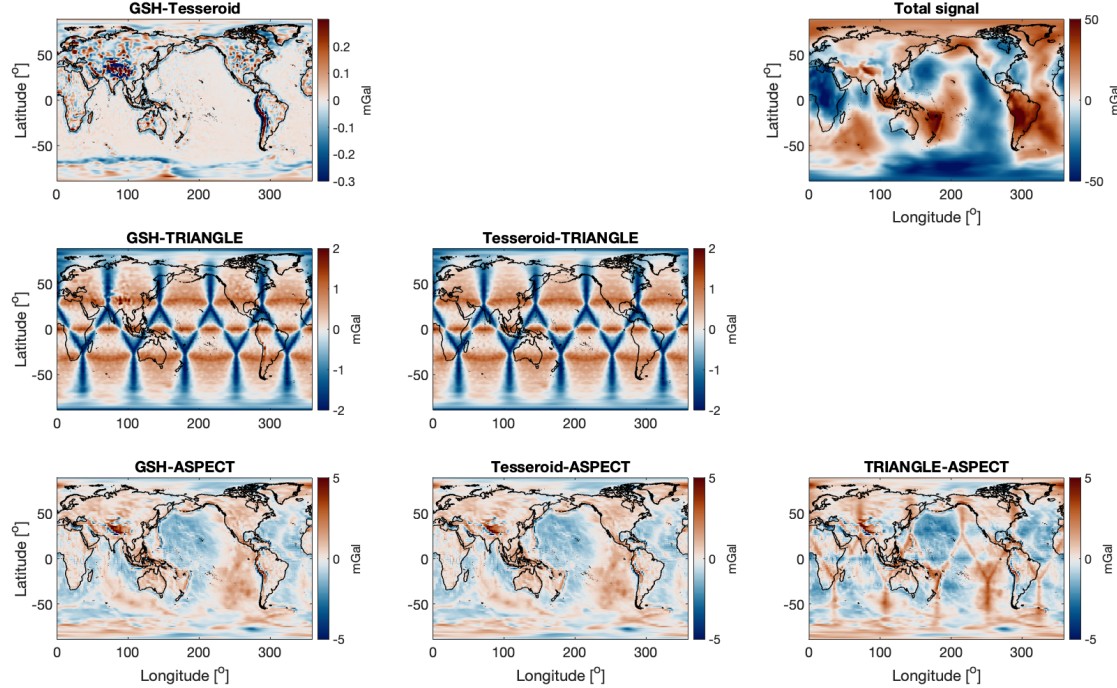

**Figure 9.** Gravity radial component at 250 km height comparison of the forward modelled WINTERC-G model by the different forward modelling approaches. Spherical harmonic coefficients 2-179 degree and order are shown.

overall similar solution of the WINTERC-G gravitational signal well within the typical uncertainty of global gravity data. GSH-ASPECT has a standard deviation of $0.83\,\mathrm{mGal}$ that indicates rather non-computational differences like data use etc. It is expected that higher resolution computations will let the solutions converge to similar precision as in the simple shell tests, 420  but that the complexity of the WINTERC-G model acquires higher resolutions settings.

| Solution (mGal) | Mean | Std. | Min | Max |
|---|---|---|---|---|
| Total signal | -0.80997 | 21.3321 | -65.7293 | 73.699 |
| GSH - Tesseroid | -0.0010525 | 0.075833 | -0.90358 | 2.3141 |
| GSH - Triangle | -0.10338 | 0.70749 | -2.4532 | 2.4901 |
| GSH - ASPECT | -0.013693 | 0.8303 | -7.6905 | 9.555 |
| Tesseroid - Triangle | -0.10233 | 0.70519 | -2.4069 | 1.7032 |
| Tesseroid - ASPECT | -0.012641 | 0.81045 | -7.0426 | 8.9018 |
| Triangle - ASPECT | 0.089689 | 1.0221 | -7.2182 | 8.071 |

**Table 4.** Statistical results from the WINTERC-G-grav benchmark.





# 5 Discussion

The gravity field is extremely sensitive to the volume of the modelled masses and therefore the exact representation of the boundaries of individual mass layers. Different gravity signatures can be computed when a modeler is not aware of this. The WINTERC-G lithosphere model is constructed with a spectral gravity forward modelling approach. This has consequences
for the inverted densities and other physical parameters, when the model is used as prior information in independent studies using different codes and gravity forward modelling/inversion approaches. This benchmark study was performed to: i) assess the differences arising from using different available gravity forward modelling approaches on a realistic global 3-D density distribution from the surface down to the base of the upper mantle (WINTERC-G model); and ii) to independently asses the reproducibility of WINTERC-G from a gravity field point of view. The different tests devised in this study are summarized in Table 5. The GSH code is able to forward model the WINTERC-G gravity signal to similar precision as the WINTERC-G

| Test | GSH | Tesseroid | Triangle | ASPECT |
|---|---|---|---|---|
| case 1: homogeneous shell 2 km (mGal) | $10^{-9}$ | 0.04 | 0.046 | 4e-6 |
| case 1: homogeneous shell 5 km (mGal) | $10^{-9}$ | 0.10 | 0.035 | 4e-6 |
| case 1: homogeneous shell 10 km (mGal) | $10^{-9}$ | 0.20 | 0.031 | <1e-6 |
| **WINTERC-G integration test** | | | | |
| WINTERC-G - XGM2016 | $\pm8.2$ m (8.2 %) | | | |
| GSH - XGM2016 | $\pm8.0$ m (8.0 %) | | | |
| WINTERC-G - GSH | $\pm6.1$ m (6.1 %) | | | |
| Comparison (standard deviation) | **Shell case 2** | **Shell case 3** | **WINTERC-G** | |
| GSH - Tesseroid | 0.055 mGal (0.3 %) | 0.027 mGal (0.02 %) | 0.076 mGal (0.36 %) | |
| GSH - Triangle | 0.215 mGal (1.2 %) | 0.755 mGal (0.5 %) | 0.707 mGal (3.3 %) | |
| GSH - ASPECT | 0.215 mGal (1.2 %) | 1.462 mGal (1.0 %) | 0.830 mGal (3.9 %) | |
| Tesseroid - Triangle | 0.225 mGal (1.3 %) | 0.754 mGal (0.5 %) | 0.7052 mGal (3.3 %) | |
| Tesseroid - ASPECT | 0.192 mGal (1.1 %) | 1.455 mGal (1.0 %) | 0.810 mGal (3.8 %) | |
| Triangle - ASPECT | 0.306 mGal (1.7 %) | 1.688 mGal (1.1 %) | 1.022 mGal (4.8 %) | |

**Table 5.** A summary of the various benchmark tests described in this paper.


model is intended. The variations with the XGM2016 gravity field data has similar variations as the WINTERC-G dedicated solution. The difference of $\pm6.1$ m in geoid between the WINTERC-G code and GSH code solutions is mostly high-wavelength signal. This spectral region is not well defined in WINTERC-G and is mostly noise. Therefore, we compared the forward modeled solutions of the four independent approaches at satellite height (250 km). This was done because of the fact that
no extra information but noise is added by downward continuation of the satellite observed gravity field. The global gravity data that was used in WINTERC-G was mostly obtained around this height. The raising of the synthesis height is suppressing shorter wavelength features in the model, and therefore the differences in the various solutions, explaining the lower relative





changes (3.0 percent). The satellite altitude also guarantees that the distance from the masses remains larger than the size of volume elements so that the forward modelling is not affected by the discretization error.

As a first 'sanity check' we used a homogeneous spherical shell to show that all codes reproduce the gravity effect of such a simple model well with an exact analytical solution (Shell test 1). The largest errors are seen with the tesseroid code, but even here the relative accuracy achieved is still on the order of $\approx 10^{-4}$. Thus, there are no significant numerical issues effecting any of the methods. A slight problem is that triangle approach artificially imprint minor spatial patterns on the predicted gravity field. When we used a slightly more intricate shell model (Shell test 2), which contains an internal density variations, the

disagreements between the methods increased. Taking the spherical harmonics code as reference, the differences are on the order of 1 percent, although tesseroids and spherical harmonics agree even better (0.3 percent). The good agreement between spherical harmonics and tesseroids compared to the two other methods highlights the impact of the parametrization: the former two methods are inherently adapted to an equi-angular grids, whereas the latter methods require some amount of interpolation. When the gravity variations are due to an undulating density interface (Shell test 3), the disagreements are lower compared

to results of Shell test 2. Tesseroids and spherical harmonics see a reduction of relative difference by a factor of 10, while the differences for triangles and ASPECT reduce only slightly. The undulating density interface is less challenging for the algorithms than the lateral density variations. Only the ASPECT residuals show similar performance between shell test 2 and 3. ASPECT is the only code that is not supporting variable layer geometries and needs interpolation from the WINTERC-G model onto an equi-thickness grid.

The result for the complete integration of the WINTERC-G model can now be interpreted due to the shell test results. Tesseroids and the spherical harmonic approach agree again very well and consistently achieve a relative agreement of 0.3 percent with each other. This corresponds to the accuracy achieved with the laterally variable density structure, so this seems to be the limit in precision between these two approaches. One caveat is the observation height of 250 km, which suppresses the short-wavelength differences. If WINTERC-G were to be used as starting model for a more regional model, integrating airborne

and ground gravity data would be an important step. However, we have not compared the two methods at or near ground level, since this is computationally unfeasible for tesseroids on a global scale, due to the needed increase of resolution to get similar precision. Triangles could be a viable choice to model WINTERC-G. However, at the resolution level 8 of the triangular refinement, the relative differences are still $\sim 3$ percent, so the resolution needs to be increased further, which is unpractical. If the accuracy of the triangle method could be further increased, it would open up interesting possibilities to directly link

gravity and seismological modelling. Seismological models are often parametrised in terms of point values, not volumes. The triangular integration provides a consistent way to associate a gravity response to these point values and would circumvent interpolation to volume elements in a joint treatment of seismological and gravity data. The ASPECT modelling approach is not viable to represent the complete WINTERC-G model at this stage. The main limitation is that the layered WINTERC-G model needs to be voxelised and the uppermost layers of WINTERC-G contain the topography (associated with a large gravity

signal), which leads to unrealistic requirements for vertical resolution. It should be noted that the voxelisation approach was acceptable in Shell test 3, where the disagreements were merely $\sim 1$ percent. However, when the entire WINTERC-G model was used, the differences from the individual layers accumulated to a final relative difference of approximate 5 percent. The





residuals of the ASPECT solution with respect to the spherical harmonics results clearly reflect water depth, topography, and crustal structures. A considerable accuracy improvement is therefore expected if the topography and bathymetry were handled independently from ASPECT.

GSH's ability to represent the lateral varying densities as much as possible makes this approach most suitable for forward modelling of global lithosphere density models. The GSH software is built for global models with lateral varying parameters, e.g. boundaries and density, but is less suitable for regional models. It is most suitable for WINTERC-G like models based on spherical harmonic basis functions to represent the gravity field. The GSH software would be less suitable for models using a spatial forward modelling approach in the inversion, like LITHO1.0 (i.e. triangles). The resolution issue is mostly related to the Nyquist criteria. So, if the information is distributed $1 \times 1$ arcdeg on a equi-angular grid, the software would only need $1° \times 1°$ deg resolution. So, increased lateral resolution is needed to improve the precision of the solution. The radial resolution is dictated by the amount of layers the GSH has to model. However, if there would be a regional (higher) information resolution, the global resolution would need to be increased to the highest resolution present in the model, increasing the computation cost. Here, spatial techniques are more favourable. Another disadvantage of spherical harmonics is modelling of lateral jumps in the boundaries. Non-removable oscillations in gravity field occur near such a high-gradient region (e.g. Himalaya). The implementation of the WINTERC-G model was very straight forward and resulted in high precision results for the GSH approach.

The tesseroid parametrization offers a great deal of flexibility, since each volume element is described explicitly. This makes tesseroids ideally suited to represent more regional geological models, which are more complicated than a simple layered structure. However, this flexibility comes at the expense of computation time because the gravity kernel needs to be evaluated for each tesseroid-station pair. This leads to a computation time scaling behaviour of $\mathcal{O}(N_s \cdot N_t)$, where $N_s$ is the number of stations and $N_t$ is the number of tesseroids. Since the number of stations and tesseroids increases rapidly if the area of investigation is enlarged or the spacing is decreased, there are limitations on the achievable resolution on a global scale. Computationally, the tesseroid calculations are limited by CPU time but require negligible RAM. The calculation is highly parallel, so the tesseroid calculation would benefit strongly from parallelization. However, a more sophisticated solution would make use of adaptive parametrizations (*Szwillus and Götze*, 2017) to improve the numerical complexity for layered models such as WINTERC-G.

Integration of triangular grids is affected by grid irregularities so that there are multiple ways to define a volume element for each node. Furthermore, there are possibly no analytical expressions for the kernel functions (here, these functions are said to relate the point of calculation with the centre of mass of each triangle as a point-wise function). In all tests except the one for the homogeneous shell, the density structure needed to be interpolated from the native equi-angular grid into the triangular grid. Besides the differences in the mass due to the triangulation itself, the differences thus include the effect of the interpolation. The best result (compared with other integration schemes) was achieved by using the spline interpolation and the 0.5 arc-deg spatial resolution, which reduced the largest triangular artifacts significantly. What appeared very important was a vertical refinement of the data. All the 13 layers spanning $400\,\mathrm{km}$ of mass from the surface downward were refined to thin slices with a thickness of 2km maximally. Handling the triangular grids rather correspond with the scattered data representation while the



integration can easily be done in parallel (here the integration was performed on an ordinary PC) and the data indexing allows for a multi-resolution approach (i.e., where possible the triangles can be divided to smaller surface/volume elements).

The ASPECT code is first and foremost a geodynamics code designed to solve the mass, momentum and energy conservation equations on massively parallel architectures. Forward gravity calculations based on Gauss-Legendre quadrature were added to it as a post-processor. Despite its relying on octree-based mesh refinement (*Burstedde et al.*, 2011) which allows to increase resolution in or around areas according to user-prescribed criteria, we found this approach to be very sensitive to density interfaces (as in the shell test 3). When adequate resolution was used the obtained, results compare favorably with the other

methods but the memory requirements as well as the computational time were found to be prohibitive as compared to other approaches showcased in this work.

The biggest issue with the ASPECT approach is the inability to accurately model a variable density interface. Codes that cannot account for variable thickness in mass layers will find the WINTERC-G model difficult to implement. This was best seen in the ASPECT results. To investigate this more, we have examined the effect constant layered-based codes and codes

using variable geometry layers with respect to their resulting gravity field solutions.

1. Representing the model with varying boundary between the crust and the mantle, approximated by the spherical harmonics functions.

2. Representing the model in equal thick layers by changing the density laterally.

The densities of the model will only have $\rho_{crust} = 2900 \ \mathrm{kg\,m^{-3}}$ and $\rho_{mantle} = 3300 \ \mathrm{kg\,m^{-3}}$ so that no interpolation is

needed. In the second approach, if the middle point of the layer is above the geometric boundary (Moho), then the density is that of the crust otherwise it will get the mantle density assigned. So, when the number of layers is increased, the radial resolution of the density interface is increased as well. This test represents a simple way to model the difference between gridded models and geometrically bounded models. The results for the gravity effect of the density interface are shown in Figure 10 by the black lines. The difference between the two solutions is largest for radial resolution of 10 km thick layers,

which is already a high radial-resolution for fully global numerical models in mantle convection studies. For example the 400 km deep WINTERC-G model is only represented by 13 layers. Differences in geoid undulations of $55 \ \mathrm{m}$ can occur, which is more than 50 percent of the observed geoid on Earth. Even with a layer thickness of $1 \ \mathrm{km}$, both approaches differ significantly. The 100 m radial resolution produces sub-meter differences. At $10 \ \mathrm{meter}$ thick layers the differences between the two models become insignificant ($\approx 10^{-11} \ \mathrm{m}$). Both of these radial resolutions are too computationally expensive when global modelling

is used. A resolution of 100 m layers for the first $400 \ \mathrm{km}$ (typical high resolution upper mantle model) would result in 4000 layers, having a lateral resolution of 0.5 degree, it means 518.4 million elements for only the upper mantle. Working with such matrices is even today very challenging.

Currently, the ASPECT code needs a equal-thickness layer grid as input file. Therefore, the WINTERC-G model needs to be converted to a grid-cube file. This is done by a python parser script (attached to the manuscript). The grid mass elements will

be calculated by taking into account the different volumes and densities in the layers of WINTERC-G. The thickness of the mass cubes can be chosen in the parser file by cutting up the WINTERC-G model in several equi-thickness layers. For this test,





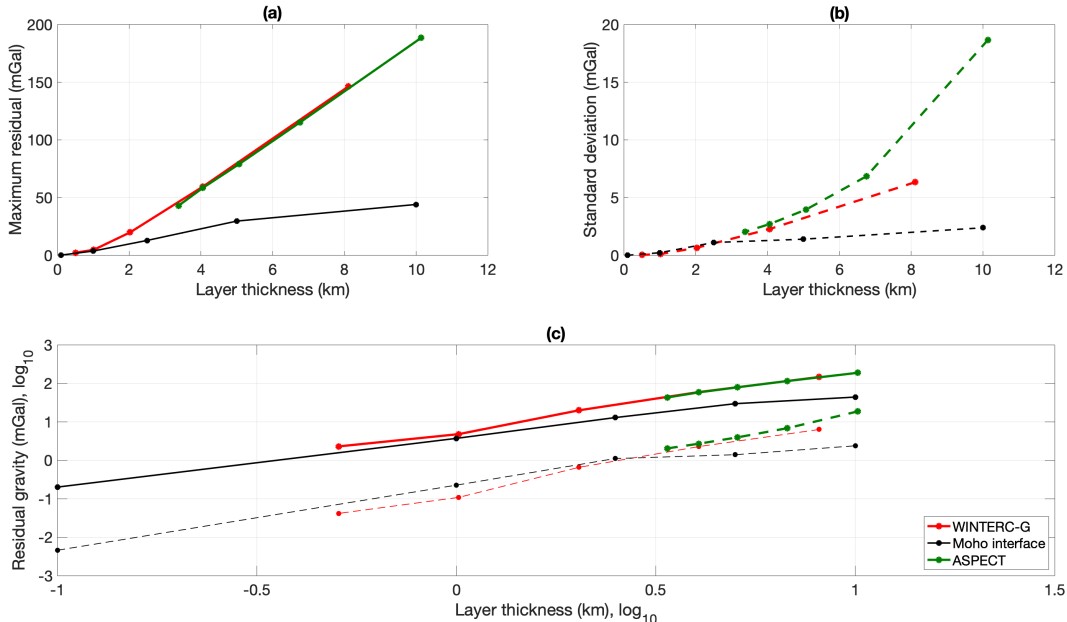

**Figure 10.** a) Maximum gravity error for equal layered model versus geometrically bounded model for the simple 80 km crust-mantle model (black) and the full WINTERC-G model by GSH (red) and ASPECT (green) b) similar to a) but the standard deviation of the residuals is plotted c) Maximum (solid lines) and standard deviation (dashed lines) residuals but plotted on a logarithmic scale.

we have investigated the gravity difference for a cube grid of 50, 100, 200, 400, and 800 layers, equivalent to layer thicknesses of 8.109, 4.054, 2.027, 1.013, and 0.506 km. These mass cube grid models are compared to the GSH solution of WINTERC-G layered model. The results are plotted in Figure 10 in the red lines. Similar to the Moho interface experiment, the WINTERC-G

model shows an increase in accuracy for small layer thickness. Radial resolution plays a role in the ASPECT solutions and for the full model is even more important than for a single density interface. The WINTERC-G model constructed in 8 km thick layers has differences of around 150 mGal (std. 6.3 mGal), whereas the crust-mantle interface has $< 50$ mGal (standard deviation of 2.4 mGal) differences from the GSH solution. Both become more accurate with the reduction of layer thickness. For layers of approximate 1 km thick, the WINTERC-G model has 4.8 mGal (standard deviation of 0.11 mGal) differences

with the GSH solution and similar results were seen in the Moho interface experiment. The highest radial resolution for the WINTERC-G model solution has layers of 506 m thick (reproducing the model in 800 individual layers). This generates a 5.4 Gb ASCII file, which is much larger than the 2 Gb limit of ASPECT input files (300 layers with a thickness of 1.35 km is around the 2 Gb limit). The difference between the GSH solution and the high resolution parser data cube is maximum 2.3 mGal (standard deviation of 0.042 mGal). So, the parser seems to converge to a similar solution as the GSH code.

However, the high resolution is currently a problem for the ASPECT code. Maybe, a newer version of ASPECT will be able





to load larger input files, but the question arises if the code will then still be practical. This high radial resolution will increase the computational effort to get an accurate solution. It would be better to assess if ASPECT would be able to adjust its mesh to the boundaries of the WINTERC-G model, which means on-equal thickness layered mesh. This is currently not yet possible in ASPECT. Bearing these limitations in mind, all four codes seem to be able to reproduce the gravity field of the complex 3D

upper mantle model well within geophysical uncertainties.

  Overall, the discussed approaches show similar precision attainable, but have several differences with respect to handle the density models, but what about the practicality of the algorithms. What is their demands on the RAM and CPU-time, how well can they be made suitable for parallel computing, and are the algorithms able to have local enhancement. The figures discussed here are orders of magnitude as the approaches have been implemented on different type of machines. Therefore, the exact

figures on CPU usage different due to the different hardware setup. However the order of magnitudes already give an indication of the performance and practicality of the different approaches.

  The run-time of the GSH approach for the full WINTERC-G model on a standard laptop is 3 minutes for the analysis of the WINTERC-G coefficients and 7 minutes for the synthesis, in total 10 minutes. The shell test run-time is negligible. The run-time of the tesseroid approach was for the complete WINTERC-G model approximately 10 hours. The simple shell test

was run within 20 minutes. The orders of magnitude of the run-time are for a full WINTERC-G model takes 0.5 day for 204 layers and 192,000 nodes (the model was vertically/laterally refined). The shell test took about 40 minutes for approximate 10 layers and 192,000 nodes and for the L7 resolution that, which is the native WINTERC-G resolution, everything is about five to ten times faster. The run-time for the WINTERC-G model was for 90+90 model on 3 cores (nq+3), 210,000 seconds i.e. 7.1days. The shell test 1 took on 1 core with 1,536 elements around 0.7 hours, for 6,144 elements around 2.8 hours, and

for 24,576 elements around 11h. This makes the current ASPECT forward modelling part the slowest of the four. The GSH approach stands out in performance and speed and is suggested to be the optimal choice for global inversion studies where, due to the multiple runs, speeds is of the essence.

  For GSH, the memory usage scales with resolution, but for the global 0.5x0.5 degree resolution of WINTERC-G this can be considered negligible (order of several megabytes.) The most demanding process is the least-squares fitting of the SH coeffi-

cients and the calculation of the Legendre function, which are related to the resolution of the model.The memory consumption of the tesseroid algorithm is negligible (on the order of a few megabytes), since the gravity effect of each tesseroid on all measurement locations is calculated individually. The triangle algorithm has relatively low demands on RAM (each calculation point is treated independently with respect to the whole input data matrix, a sort of vectorisation); nothing serious for 0.5 arc-deg resolution. However, RAM can be limited in the pre-integration phase if naive algorithms are used for search/indexing.

Large matrices can occur (example, if col/rows of the matrix is dedicated to all data points, and when finding neighbor points (all to all search), it might create matrix sizes of 50,000x50,000 elements for L7 and 190,000x190,000 elements for L8. This is not an issue for the users but for the developers of the grids (or those who inspect them). In the current version of the ASPECT code, memory consumption is a bottleneck. The input density grid file from the python parser is currently limited by 2 Gb by the ASPECT code. This protects the user to have enormous memory consuming runs with the code, but it limits the resolution



of the density model inserted into ASPECT. Especially for lateral varying density interfaces this proves to be a large source of erroneous gravity solutions.

The GSH computation can be performed in parallel with respect to the amount of layers. The layers are independent from each other and the corresponding coefficients are added to obtain the total SH coefficients of the model. However, lateral selected or regional modelling is not possible as the GSH needs global information of the layers density distribution and its 595 geometry. The least square fitting could be performed in? parallel with proper numerical toolboxes. Local enhancement is not possible for GSH, which is one of the biggest drawbacks of the GSH code. Only an increase of the amount of spherical harmonic coefficients would improve the resolution of the gravity output. For regional studies with high resolution, spatial forward modelling approaches are then advised. Parallelisation of the tesseroid code would be straightforward. Increasing the resolution adaptable is also straightforward when tesseroids are combined with hierarchical subdivision methods like quadtrees 600 (e.g. *Szwillus and Götze* (2017)). In this manner, speed-ups without significant loss of accuracy can be achieved. For triangle approach it is also easy to make the forward problem parallel, the integration loop that runs over all the calculation points can be split into more "segments". It can be have easy vectorisation, but for the inverse problem the same drawbacks are envisioned like for other approaches. Local enhancements are possible; large triangles can be replaced with smaller ones. For the triangle approach practically, enhancement means to remove one column per triangle element and adding few more for 605 the smaller triangles in the input file. A limitation is as said, it is crucial to perform an analysis before the grid is integrated. After each enhancement the grid (or its part) must be reanalysed (and the indexing/data ordering appropriately updated) to get correct surface/volume elements (these numbers cannot come from the analytical formulas, they have to be estimated because the triangle grids are irregular. Parallelisation is integral part of the ASPECT code since the mesh is partitioned across all processors. ASPECT is ideal for local enhancements, as it relies on adaptive mesh refinement so that specific areas, geophysical 610 features or material interfaces can be more densely meshed. Mainly the lateral varying density interfaces prove difficulty for the ASPECT code. This would mean that if global density models of the Earth would not use lateral varying density interface, but instead use a more grid-type format all four codes would perform almost equally well both in precision as in practicality.

## 6 Conclusions

This benchmark study is focused on the computation of the gravitational potential field associated with the crustal and upper 615 mantle model WINTERC-G (*Fullea et al.*, 2020). Four independent forward modelling approaches/codes are tested against the WINTERC-G-based spectral forward modelling code used in the inversion. The four codes differ in the methodology assumed: a global spherical harmonic solution (GSH code) *Root et al.* (2016), a tesseroid based code (*Uieda et al.*, 2016), an integration on triangle volume elements (*Sebera et al.*, 2018), and a hexacon-based code inside the open-source software ASPECT (*Kronbichler et al.*, 2012; *Heister et al.*, 2017). The GSH, TESS, TRI, and ASP codes are able to reproduce gravity 620 fields that are significantly similar to WINTERC-G solution. WINTERC-G is successfully tested to check the recovery of the input gravity data used to constrain it: XGM2016. Among all four tested codes, ASPECT has more difficulty in computing the correct gravity solution, when radially varying density contrasts are present.





Simple shell tests shows that all four codes can produce similar gravitational potential fields, suitable for modelling of satellite-acquired gravity data. The differences between the forward modelling schemes are all below 1.5 percent of the modeled signal, in which the tesseroid and GSH codes produced the most similar results ($<$0.3 percent). The biggest issue for the triangle code is the characteristic pattern in the residuals that illustrates the grid selection. Triangles provide a realistic gravity signal with RMS$< 1$ mGal but compared to GSH/tesseroids these residuals are about 10 times larger. Increasing the resolution and filtering is capable of removing these imprints to some extent but major issue is use of point-wise kernel values along with irregular grid on the sphere (there is no perfectly uniform triangular grid on the sphere). The ASPECT-based code performs worst in the simple shell tests, especially in the forward modelling of the gravity signature of a density contrast of a depth-varying boundary.

The GSH code shows that it can produce almost similar potential fields to the internal spectral code that was used in the development of the WINTERC-G model. Mainly short wavelength noise is seen between both forward modelling codes, that can be attributed to the different way the spherical harmonic analysis of the varying boundaries of the mass layer is performed. This produces small differences especially at high gradient values of the boundary variations, introducing mostly short wavelength differences. The spatial forward modelling schemes still have difficulty in reproducing similar gravity field solutions and would have to go to unrealistic high resolutions, resulting in enormous computation efforts. Care must be taken with any forward modelling software as the approximation of the geometry of the WINTERC-G model may deteriorate the gravity field solution, if the density parametrization of this model is not taken into account.

*Code and data availability.* The software and model that are used in the study are all open-source. They can be found at the following locations:

- The GSH approach is available at https://github.com/bartroot/GSH.
- The Tesseroid approached used the open source package Tesseroids, which is available at https://github.com/leouieda/tesseroids/.
- The Matlab codes for the integration of the triangle grid is available in the supplementary material.
- ASPECT is an open-source software project and can be obtained at https://aspect.geodynamics.org/. Version 2.4.0-pre was used.
- Spectral code (STOPOC) that was used in the development of WINTERC-G is attached to the supplementary material.
- The WINTERC-G model that was used in this comparison can be found in *Fullea et al.* (2021).
- The python parser to change the layered WINTERC-G model into a density cube for the ASPECT implementation is added to the supplementary material.

## Appendix A: Mathematical description of the forward modelling code used in the construction of WINTERC-G

The inversion code used to construct WINTERC-G model relies on spherical harmonic forward gravity modelling code (*Fullea et al.*, 2020). The approach is based on the derivation of Stokes' potential coefficients of a 3D density layer with non-spherical boundaries. The aim of this section is to derive the formulae for computing the external gravitational field generated by a





mass layer of a 3D density distribution bounded by non-spherical boundaries with the geocentric radii $r = a(\Omega)$ and $r = b(\Omega)$.
Here, $\Omega$ stands for co-latitude and longitude, $\Omega \equiv (\vartheta, \varphi)$. We assume that the two boundaries do not intersect each other, i.e.,
$a(\Omega) \neq b(\Omega)$ for any $\Omega$. We consider, for instance, that $a(\Omega) < b(\Omega)$, i.e $a(\Omega)$ and $b(\Omega)$ are the bottom and top boundaries of
the layer, respectively.

Let the mass density $\varrho(r, \Omega)$ above the boundary $a(\Omega)$ be $\varrho_a(\Omega)$ and below the boundary $b(\Omega)$ be $\varrho_b(\Omega)$. Mathematically,

$$\lim_{r \to a^+} \varrho(r, \Omega) = \varrho_a(\Omega),$$
$$\lim_{r \to b^-} \varrho(r, \Omega) = \varrho_b(\Omega). \tag{A1}$$

Let $\varrho(r, \Omega)$ inside the layer change linearly with the radius $r$ of a mass-density point, i.e.,

$$\varrho(r, \Omega) = \alpha(\Omega)r + \beta(\Omega) \tag{A2}$$

for $a(\Omega) \leq r \leq b(\Omega)$. Functions $\alpha(\Omega)$ and $\beta(\Omega)$ are given by the boundary density values $\varrho_a(\Omega)$ and $\varrho_b(\Omega)$:

$$\alpha(\Omega) = \frac{\varrho_b(\Omega) - \varrho_a(\Omega)}{b(\Omega) - a(\Omega)}, \qquad \beta(\Omega) = \varrho_a(\Omega) - \alpha(\Omega)a(\Omega). \tag{A3}$$

Let us now compute the gravitational potential $V$ induced by the mass-density layer,

$$V(r, \Omega) = G \int_{\Omega_0} \int_{r'=a(\Omega')}^{b(\Omega')} \frac{\varrho(r', \Omega')}{L(r, \psi, r')} r'^2 \, dr' \, d\Omega', \tag{A4}$$

where $G$ is the Newton's gravitational constant, $\Omega_0$ is the full solid angle, $d\Omega' = \sin\vartheta' d\vartheta' d\varphi'$, $L(r, \psi, r')$ is the spatial distance
between the computation point $(r, \Omega)$ and an integration point $(r', \Omega')$,

$$L(r, \psi, r') := \sqrt{r^2 + r'^2 - 2rr' \cos\psi}, \tag{A5}$$

and $\psi$ is the angular distance between geocentric directions $\Omega$ and $\Omega'$. For $r > r'$, the reciprocal distance $1/L$ can be expanded
into a uniformly convergent series of Legendre polynomials,

$$\frac{1}{L(r, \psi, r')} = \frac{1}{r} \sum_{j=0}^{\infty} \left(\frac{r'}{r}\right)^j P_j(\cos\psi). \tag{A6}$$

Using the Laplace addition theorem (*Varshalovich et al.*, 1989), the potential $V$ at the external point $(r, \Omega)$, that is for $r > R > b(\Omega)$, can be expressed in terms of solid spherical harmonics,

$$V(r, \Omega) = \frac{GM}{R} \sum_{j=0}^{j_{max}} \sum_{m=-j}^{j} \left(\frac{R}{r}\right)^{j+1} V_{jm} Y_{jm}(\Omega), \tag{A7}$$

where $M$ is the mass of the Earth and $Y_{jm}(\Omega)$ are the fully normalized scalar spherical harmonics of degree $j$ and order
$m$, respectively (*Varshalovich et al.*, 1989). The factor $GM/R$ is used to express the potential $V$ with respect to the mean





gravitational potential of the Earth. Consequently, the potential coefficients $V_{jm}$ are normalized by the average density of the Earth, $\varrho_{\text{mean}}$, such that

$$V_{jm} = \frac{3}{\varrho_{\text{mean}}} \frac{\sigma_{jm}}{2j+1}. \tag{A8}$$

The scaled potential coefficients $\sigma_{jm}$ express the contributions of the various mass density distributions inside the Earth to the external gravitational field. In the case of a mass-density layer with the density $\varrho(r,\Omega)$ bounded by surfaces $r = a(\Omega)$ and $r = b(\Omega)$, the potential coefficients $\sigma_{jm}$ are

$$\sigma_{jm} = \int\limits_{\Omega_0} \int\limits_{r'=a(\Omega')}^{b(\Omega')} \varrho(r',\Omega') \left(\frac{r'}{R}\right)^{j+2} Y^*_{jm}(\Omega') dr' d\Omega', \tag{A9}$$

where the asterisk denotes the complex conjugation. Substituting for $\varrho(r',\Omega')$ from (A2) into (A9) and integrating the result with respect to $r'$ gives

$$\sigma_{jm} = \frac{1}{R^{j+3}} \int\limits_{\Omega_0} \left\{ \frac{\alpha(\Omega')}{j+4} \left( [b(\Omega')]^{j+4} - [a(\Omega')]^{j+4} \right) + \frac{\beta(\Omega')}{j+3} \left( [b(\Omega')]^{j+3} - [a(\Omega')]^{j+3} \right) \right\} Y^*_{jm}(\Omega') d\Omega', \tag{A10}$$

In view of the last expression, it is convenient to express the boundary topographies $a(\Omega)$ and $b(\Omega)$ in the form

$$
\begin{aligned}
a(\Omega) &= R_a + s(\Omega), \\
b(\Omega) &= R_b + t(\Omega),
\end{aligned} \tag{A11}
$$

where $R_a$ and $R_b$ are mean radii of $a(\Omega)$ and $b(\Omega)$, and $s(\Omega)$ and $t(\Omega)$ are undulations of $a(\Omega)$ and $b(\Omega)$ with respect to the mean radii. The power of the boundary radii in (A10) will now be expressed in terms of spherical harmonics (*Martinec et al.*, 1989; *Fullea et al.*, 2015). For integer $n$, $n \geq 1$, and using (A11), the $n$th power of the topography $a(\Omega)$ can be written as a power series of $s(\Omega)/R_a$ by means of the binomial theorem:

$$
\begin{aligned}
[a(\Omega)]^n &= R_a^n \left[ 1 + \frac{s(\Omega)}{R_a} \right]^n \\
&= R_a^n \sum_{k=0}^{n} \binom{n}{k} \left[ \frac{s(\Omega)}{R_a} \right]^k.
\end{aligned} \tag{A12}
$$

A similar expansion holds for $[b(\Omega)]^n$. Substituting (A12) into (A10) and introducing the ratios

$$p_a = \frac{R_a}{R}, \qquad p_b = \frac{R_b}{R}, \tag{A13}$$

the final expression for the scaled potential coefficients is

$$\sigma_{jm} = \int\limits_{\Omega_0} \left[ \alpha(\Omega') \frac{R}{j+4} \sum_{k=0}^{j+4} \binom{j+4}{k} \left\{ p_b^{j+4} \left[ \frac{t(\Omega')}{R_b} \right]^k - p_a^{j+4} \left[ \frac{s(\Omega')}{R_a} \right]^k \right\} \right.$$





$$+\beta(\Omega')\frac{1}{j+3}\sum_{k=0}^{j+3}\binom{j+3}{k}\left\{p_b^{j+3}\left[\frac{t(\Omega')}{R_b}\right]^k-p_a^{j+3}\left[\frac{s(\Omega')}{R_a}\right]^k\right\}\right]\right]Y_{jm}^*(\Omega')d\Omega'. \tag{A14}$$

In a particular case, when the both bounding topographies are spherical, $s(\Omega)=t(\Omega)=0$ for any $\Omega$, the only non-zero contributions to the integrand in (A14) are for $k=0$, and (A14) reduces to

$$\sigma_{jm}=\int_{\Omega_0}\left[\alpha(\Omega')\frac{R}{j+4}(p_b^{j+4}-p_a^{j+4})+\beta(\Omega')\frac{1}{j+3}\left(p_b^{j+3}-p_a^{j+3}\right)\right]Y_{jm}^*(\Omega')d\Omega'. \tag{A15}$$

If, in addition, the density does not change in the layer in radial direction, $\varrho_b(\Omega)=\varrho_a(\Omega)$. Then, (A3) implies that $\alpha(\Omega)=0$, $\beta(\Omega)=\varrho_a(\Omega)$ and (A15) further reduces to

$$\sigma_{jm}=\int_{\Omega_0}\frac{\varrho_a(\Omega')}{j+3}\left(p_b^{j+3}-p_a^{j+3}\right)Y_{jm}^*(\Omega')d\Omega'. \tag{A16}$$

The last expression gives the solution for a spherical shell of constant thickness, but with lateral varying densities $\varrho_a(\Omega')$.

## Appendix B: Effect of triangle area on the integration

Since there is no spherical triangular grid with constant-area surface elements, the spherical triangles necessarily differ in the sides and angles. When assigning an area to a node according to Figure 1, there are multiple options. We have studied five different options using the spherical shell:

1. Constant size - each node is given the same area that is proportional to a number of points on the sphere

2. Local simple average - a node is given an average area value estimated with the neighbouring triangles (see Figure 1)

3. Weighted local average - similar to previous but the neighboring triangles are weighted depending on the magnitude of the inner angle or its sine

4. Sum of thirds - each node is given an area equal to a sum of thirds from surrounding triangles

5. Centre of mass - each node is given the area according to Figure 1 - a sum of smaller spherical triangles delimited by the centres of mass and the side midpoints.

The results for the spherical shell using these settings is discussed in terms of Table B1. Although options 3-5 nearly reach the exact surface/volume of the sphere ($4\pi$), the gravity residuals compared with Equation (4) are higher than for option 2 (local average). This is due to the fact that the irregular triangular grid always produces triangular patterns and these cannot be reduced just by using more accurate surface/volume elements. Thus, the most suitable option based on the shell test seems to be option 2 even though only 5 significant digits of the total surface is preserved. Note there is a two orders magnitude




| Option | Total sph. area | RMS of residuals (mGal) |
|---|---|---|
| Sph. shell | 12.56637061435917 | - |
| 1 | 12.56637061435917 | 10.9 |
| 2 | 12.56622208332847 | 0.08 |
| 3 | 12.56637061439066 | 0.72 |
| 4 | 12.56637061439047 | 0.46 |
| 5 | 12.56637061423496 | 0.46 |

**Table B1.** Surface element options in terms of the total spherical area and gravity residuals. For option 1 the total area is exact since the triangle area is calculated directly from $4\pi$.

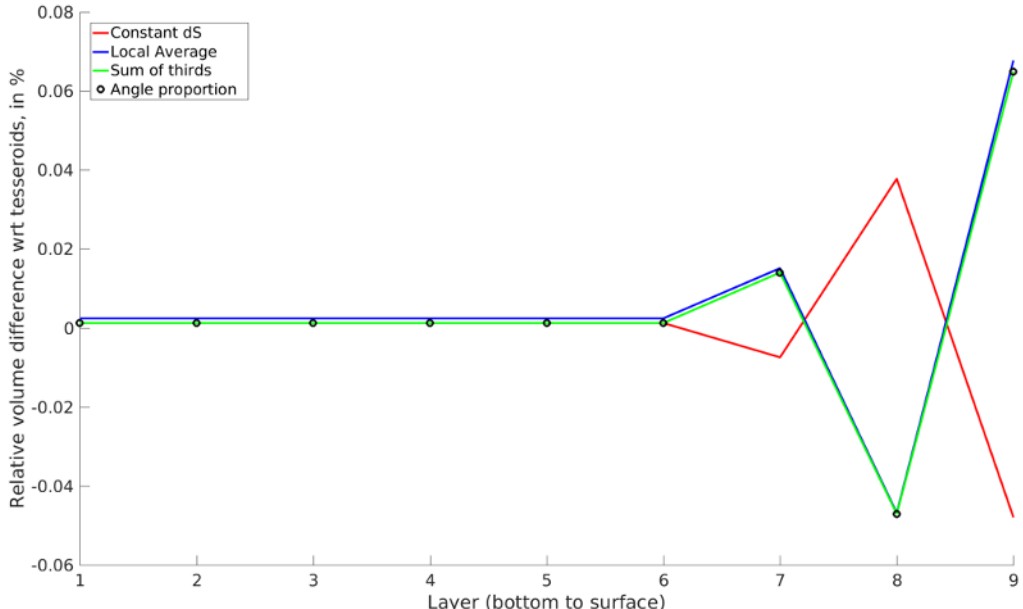

**Figure B1.** Relative volume differences per layer (from 400 km) of WINTERC-G with respect to the tesseroids.

improvement when options 1 and 2 are compared and if possible constant surface/volume elements should not be used along
with the triangular grids.

The differences in the volume with respect to tesseroids are provided in Figure B1. The rougher topography of the layer the larger differences between the two integration schemes can be seen (typically the layers closer to the Earth's surface). However, the accuracy of the volume elements seems to be a less important driver of the triangular patterns as already indicated by the shell test - even near-exact volume elements do not help with reducing the large-scale triangular patterns (even if mean kernels
are used). Possibly, the largest artifacts come from the grid irregularity for a given 2 arc-deg spatial resolution. These large triangular patterns are basically an amplified gravity signal from close source points - the triangles are smaller than some local average so the effect of its nodes is larger (or vice versa for the opposite situation).



## Appendix C: WINTERC-G layering

The layering of WINTERC-G can be viewed in Table C1.

| Name layer | top boundary | bottom boundary | corresponding density |
|---|---|---|---|
| **Varying thickness layers** | | | |
| Water | Top continental ETOPO2 | Top ice ETOPO2 | 1030 kg/m$^3$ (constant vertically) |
| Ice | Top ice ETOPO2 | Top Bedrock ETOPO2 | 910 kg/m$^3$ (constant vertically) |
| Crust | Top Bedrock ETOPO2 | Moho (Crust1.0) | defined in file: rho_c_out.xyz (constant vertically) |
| Mantle 1 | Moho (Crust1.0) | Max. depth: 20 km | defined in file: rho_submoho_out.xyz (top) |
| | | | defined in file: rho_20km_out.xyz (bottom) |
| Mantle 2 | Max. depth: 20 km | Max. depth: 36 km | defined in file: rho_20km_out.xyz (top) |
| | | | defined in file: rho_36km_out.xyz (bottom) |
| Mantle 3 | Max. depth: 36 km | Max. depth: 56 km | defined in file: rho_36km_out.xyz (top) |
| | | | defined in file: rho_56km_out.xyz (bottom) |
| Mantle 4 | Max. depth: 56 km | 80 km depth | defined in file: rho_56km_out.xyz (top) |
| | | | defined in file: rho_80km_out.xyz (bottom) |
| **Constant thickness layers** | | | |
| Mantle 5 | 80 km | 110 km | defined in file: rho_80km_out.xyz (top) |
| | | | defined in file: rho_110km_out.xyz (bottom) |
| Mantle 6 | 110 km | 150 km | defined in file: rho_110km_out.xyz (top) |
| | | | defined in file: rho_150km_out.xyz (bottom) |
| Mantle 7 | 150 km | 200 km | defined in file: rho_150km_out.xyz (top) |
| | | | defined in file: rho_200km_out.xyz (bottom) |
| Mantle 8 | 200 km | 260 km | defined in file: rho_200km_out.xyz (top) |
| | | | defined in file: rho_260km_out.xyz (bottom) |
| Mantle 9 | 260 km | 330 km | defined in file: rho_260km_out.xyz (top) |
| | | | defined in file: rho_330km_out.xyz (bottom) |
| Mantle 10 | 330 km | 400 km | defined in file: rho_330km_out.xyz (top) |
| | | | defined in file: rho_400km_out.xyz (bottom) |

**Table C1.** The WINTERC-G layering structure used in this study.

*Author contributions.* Bart Root is the main contributor to the manuscript, drafted initial version, responsible for the GSH results and discussions, and performed the comparison of all approaches and tests. Josef Sebera was initiator of the benchmark study, responsible for the TRIANGLE approach results and discussions, and reviewed the full manuscript, Wolfgang Szwillus co-initiator of the benchmark study, responsible for the Tesseroid results and discussions, wrote the python parser, and reviewed the manuscript. Cedric Thieulot was involved





in finalizing the benchmark study, responsible for the ASPECT results and discussions, and substantially reviewed the manuscript. Zdeněk
Martinec was involved in the initial benchmark study and developed the gravity code of the WINTERC-G model, wrote appendix A, and
reviewed the manuscript. Javier Fullea is the developer of the WINTERC-G model and helped in the benchmark of all the forward modelling
codes, benchmark the parellel version of the WINTERC-G spectral code, wrote appendix C, and reviewed the manuscript.

*Competing interests.*  The authors declare that they have no conflict of interest.

*Acknowledgements.*  This work has been done in the framework of the project "3D Earth - A Dynamic Living Planet" funded by ESA as
a Support to Science Element (STSE). We would like to thank J/"org Ebbing and Roger Haagmans for their vital discussions on the topic.
Perceptually uniform colour maps were used in this study to prevent visual distortion of the data (http://www.fabiocrameri.ch/).





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
