# Peer review of "Benchmark forward gravity schemes: the gravity field of a realistic lithosphere model WINTERC-G"

_Solid Earth, 2021_

## Author Response (AR1)

Dear editor,

We thank you for considering this manuscript for publishing after addressing the comments made by the two reviewers and yourself. We will address them in chronological order. We hope that we have addressed them accordingly.

With kind regards,
Bart Root, Josef Sebera, Wolfgang Szwillus, Cedric Thieulot, Zdenek Martinec, and Javier Fullea.

To address Reviewer 1:

We would like to thank reviewer 1, Mikhail, for the kind words and summary given on our work. We are glad that you find the work satisfactory and are happy to read that you find the manuscript worthy of publication. Thank you for taking the time to review our work.

To my point, this is a very important methodological paper, which will be cited in many following studies of the gravity field. It is very clearly written and each, even small, detail is explained and documented by numerical examples. Therefore, I recommend publication of the manuscript in its present form. I have the only recommendation to perform a similar analysis for calculation of the gravity gradients, which are often used after appearance of the direct GOCE data, however, this could be a topic for the next paper.

We agree that gravity gradients, also need to be benchmarked. We are considering a future publication on this in line with ESA's 3D Earth project deliverables. A dedicated book chapter will be about benchmarking satellite gravity gradients. We will make sure to connect that chapter to this study and the related codes. For this publication we feel that it would become to cluttered with results and therefore we decided to stick to the components presented in the manuscript.

To address Reviewer 2:

Likewise, we thank reviewer 2 for the kind words and summary of our work. We appreciate the time you took to review our work and are pleased that you find it worth publishing. Thank you.

1) The benchmarks involving WINTERC-G based density structures show two different comparisons : the WINTERC-G forward modelling code is compared to the spherical harmonics code using the geoid signals (this corresponds to the data used to constrain the WINTERC-G density structure), and then the spherical harmonics code is compared to the three local integration codes using a different observable, namely the radial gravity at GOCE satellite height. Why not using the same observable for all the comparisons ?

The reason for using the geoid for the spherical codes and r-component results for the 4 other methodologies is due to the fact that the internal ASPECT code is not yet able to perform geoid calculations Thus, we have used two quantities.

2) The authors have chosen geophysical settings for the sources in the benchmarks. Maybe it could be interesting to also consider purely synthetic settings, involving localized or oscillating sources at different spatial resolutions, to show how the different algorithms perform on elementary-type of sources with a controlled spatial resolution? Rather than the relatively smooth radial gravity at satellite altitude, you could also consider ground gravity as a high-resolution observable ? However, the manuscript already comprises a whole set of logically organized examples with a detailed discussion of the results, which is valid enough to me – so this is just a suggestion, this could also be part of another paper, please decide yourself.

Thank you for extra suggestions for the benchmark. We will take this consideration with us for a future publication. In this manuscript, we wanted to show the validity of the gravity signal of WINTERC-grav for different forward modelling methodologies for the global lithospheric (deep Earth) models. Therefore, we also selected to plot the signal at satellite height as WINTERC-grav used the global data from satellite observations. We agree that an even more thorough benchmark is needed to see all differences between the codes. We are considering this for a book chapter publication on further benchmarking the code.

To address the editor:

We appreciated the time and effort you took in improving our manuscript

- you already provide all the different codes, which is great. I think it would be very useful if you could also provide the input and output data of the shell tests as a data set (on zenodo, or as a supplement). The reason I am suggesting this: If someone wants to benchmark their own gravity code, they can use your test cases as an example and compare how well their own code performs against them. This would increase the impact of your paper.

I have added the tests as a dataset in 4TU.researchdata database, where the GSH code is also uploaded. It is currently under review of receiving a doi. When this is the case we will add this doi to the publication.

- Figure 5: I would suggest to increase the size of the labels. They are quite hard to read. In addition, could you clarify at what depth this is for (a)?

We have added the depth of the density layer used in that test from WINTERC-grav. We have increased the labels and the size of figure 5 to enhance the readability of the graph.

- In section 4.2, you mention different versions of the computational approaches. So I think for reasons of reproducibility, it would be great if you could provide version numbers or commit

hashes or something along those lines for all the codes you used in your code availability statement.

We have added a statement that the previous version off the GSH code is not publicly available and was a beta.version. Furthermore, we provided a DOI link to the GSH version used in this work.

- Paragraph starting line 567: It's great that you provide computational times. But I was confused which of the times provided correspond to which method. It would be great if you could make that clearer.

We have made improvement to this paragraph, such that it is clear which forward methodology is discussed concerning the run-time.

While reading through the manuscript, I also found some typos you might want to fix:
Line 16: The Spherical harmonic basis functions produce**s** → produce
Line 29: One of the latest global gravity field model → model**s**
Line 33: such like → such as
Line 39: consisting of 1-D stage → consisting of **a** 1-D stage
Line 55: at location → at location **P**
Line 76: ranging from simple shell tests to a **the** more complex upper mantle mode --> remove "the"
Line 115: had convergence**s** issues → had convergence issues
Line 159: shows more smooth transitions, than the isosahedron grid → remove comma
Line 200: We place spherical shell at a mean depth of 100 km with respect to the Earth's 6371 km reference sphere and modelling different thicknesses of 2, 5, and 10 km. → We place a spherical shell at a mean depth of 100 km with respect to the Earth's 6371 km reference sphere and model different thicknesses of 2, 5, and 10 km.
Line 233: The mass shell in this scenario consists is described by → remove consists
Line 254: Figure 6 visualises the differences of **various the** forward modeling results → Figure 6 visualises the differences of the various forward modeling results
Line 268: this is not everywhere obvious → this is not obvious everywhere
Line 293: the locations where the Moho boundary changes much → do you mean: changes abruptly?
Line 312: which related to the limited radial resolution → which is related to the limited radial resolution
Line 331: A series of layers are defined → A series of layers is defined
Line 358: are mainly long-wavelength variations of around ±2 m are similar → are mainly long-wavelength variations of around ±2 m that are similar
Line 379: reflects theeffect → reflects the effect
Line 380: The difference are smaller → The difference**s** are smaller

We have addressed our typo's accordingly and produced a change record to show them.

Furthermore:

From our initial submission we received some feedback from the editorial team.

We have tried to revise our reference list to be in accordance with the journal's standards.

We have revised figures 1 and 10 and corresponding caption and text such that the figures are more suitable for color vision deficiencies.